# Deep Dive into Hydrologic Simulations at Global Scale: Harnessing the Power of Deep Learning and Physics-informed Differentiable Models (δHBV-globe1.0-hydroDL)

Dapeng Feng[1,2,3], Hylke Beck[4], Jens de Bruijn[3,5], Reetik Kumar Sahu[3], Yusuke Satoh[6], Yoshihide Wada[7], Jiangtao Liu[1], Ming Pan[8], Kathryn Lawson[1], Chaopeng Shen[1*]

[1] Civil and Environmental Engineering, Pennsylvania State University, University Park, PA, USA

[2] Earth System Science, Stanford University, Stanford, CA, USA

[3] Water Security Research Group, International Institute for Applied Systems Analysis (IIASA), Laxenburg, Austria

[4] Climate and Livability Initiative, Physical Science and Engineering Division, King Abdullah University of Science and Technology, Thuwal, Saudi Arabia

[5] Institute for Environmental Studies (IVM), Vrije Universiteit Amsterdam, Amsterdam, Netherlands

[6] Moon Soul Graduate School of Future Strategy, Korea Advanced Institute of Science and Technology, Daejeon, Republic of Korea

[7] Biological and Environmental Science and Engineering Division, King Abdullah University of Science and Technology, Thuwal, Saudi Arabia

[8] Center for Western Weather and Water Extremes, Scripps Institution of Oceanography, University of California San Diego, La Jolla, CA, USA

*Correspondence to*: Dapeng Feng (dpfeng@stanford.edu) Chaopeng Shen (cshen@engr.psu.edu)

**Abstract.** Accurate hydrologic modeling is vital to characterizing how the terrestrial water cycle responds to climate change. Pure deep learning (DL) models have shown to outperform process-based ones while remaining difficult to interpret. More recently, differentiable, physics-informed machine learning models with a physical backbone can systematically integrate physical equations and DL, predicting untrained variables and processes with high performance. However, it was unclear if such models are competitive for global-scale applications with a simple backbone. Therefore, we use - for the first time at this scale - differentiable hydrologic models (full name δHBV-globe1.0-hydroDL, shortened to δHBV here) to simulate the rainfall-runoff processes for 3753 basins around the world. Moreover, we compare the δHBV models to a purely data-driven long short-term memory (LSTM) model to examine their strengths and limitations. Both LSTM and the δHBV models provide competent daily hydrologic simulation capabilities in global basins, with median Kling-Gupta efficiency values close to or higher than 0.7 (and 0.78 with LSTM for a subset of 1675 basins with long-term discharge records), significantly outperforming traditional models. Moreover, regionalized differentiable models demonstrated stronger spatial generalization ability (median KGE 0.64) than a traditional parameter regionalization approach (median KGE 0.46) and even LSTM for ungauged region tests across continents. Nevertheless, relative to LSTM, the differentiable model was hampered by structural deficiencies for

cold or polar regions, and highly arid regions, and basins with significant human impacts. This study also sets the benchmark
for hydrologic estimates around the world and builds foundations for improving global hydrologic simulations.

**Short Summary.** Accurate hydrologic modeling is vital to characterizing water cycle responses to climate change. For the
first time at this scale, we use differentiable physics-informed machine learning hydrologic models to simulate rainfall-runoff
processes for 3753 basins around the world and compare them with purely data-driven and traditional modeling approaches.
This sets a benchmark for hydrologic estimates around the world and builds foundations for improving global hydrologic
simulations.

**Key Words.** Physics-informed machine learning; Differentiable hydrologic models; Global hydrologic modeling; high
resolution evaluation; Parameter regionalization; Prediction in ungauged regions
**1. Introduction**
Hydrologic models are vital tools to model and elucidate the terrestrial water cycle, and have been widely used in flood
forecasting (Maidment, 2017), water resources management (Jayakrishnan et al., 2005), and assessing climate change impacts
(Hagemann et al., 2013). Recently, deep learning (DL) models have demonstrated superior performance compared to
traditional process-based hydrologic models in accurately predicting different components of the hydrologic cycle (Shen,
2018), such as soil moisture (Fang et al., 2017, 2019; Fang and Shen, 2020), streamflow (Feng et al., 2020; Konapala et al.,
2020; Kratzert et al., 2019b; Liu et al., 2024), snow water equivalent (Cui et al., 2023; Song et al., 2024b), groundwater
(Wunsch et al., 2021) and water quality (Hansen et al., 2022; Rahmani et al., 2021; Saha et al., 2023; Song et al., 2024a; Zhi
et al., 2021). Long short-term memory (LSTM) networks, which are a type of recurrent neural network (Hochreiter and
Schmidhuber, 1997), and Transformers (Vaswani et al., 2017) are currently popular DL algorithms for handling time series
dynamics in hydrology, while other architectures can also be employed. LSTM models have established state-of-the-art
accuracy for streamflow prediction at continental and smaller scales (Feng et al., 2020, 2021; Kratzert et al., 2019a, b; Lees et
al., 2021; Mai et al., 2022).

Although DL models have shown great prediction accuracy compared to traditional models, they usually do not possess clear
physical constraints inside the model and are often considered to be "black boxes", despite recent efforts shed by some
interpretive efforts (Lees et al., 2022). Thus, purely data-driven models are limited in that they cannot predict unobservable or
untrained physical variables, which impedes the investigation of the physical relations of different hydrologic variables behind
the change in the target variable. They may also become overfitted and acquire incorrect sensitivities to inputs (Reichert et al.,
2024). In contrast, traditional process-based hydrologic models following physical laws like mass balances can provide a full
set of diagnostic outputs for hydrologic variables like soil water storage, groundwater recharge, evapotranspiration and snow
water equivalent, even though they are usually only calibrated on discharge observations (Burek et al., 2020; Müller Schmied
et al., 2014). The multivariate output nature of these models provides an opportunity for calibration on one or more observable
variables to better predict other, perhaps unobservable, variables (in reality, whether this is the case or not depends on if the
issue of parameter non-uniqueness is addressed). However, it seems quite difficult for the traditional physical model to
approach the performance level of the DL models in daily hydrograph metrics (Feng et al., 2020; Kratzert et al., 2019b) or to
improve in generalization with increasing training data (Tsai et al., 2021). In addition, traditional calibration is typically done
site-by-site and can be time- and labor-intensive. Therefore, it logically follows that integrating DL and process-based models
might enable harnessing their respective strengths while circumventing their weaknesses (Shen et al., 2023).

By combining a physical model with a DL model, differentiable modeling (Feng et al., 2022; Shen et al., 2023) provides a
systematic solution to leveraging the strengths of both model types while circumventing their limitations. In differentiable
models, we use process-based models as a backbone and insert neural networks to either provide parameters (Tsai et al., 2021)
or process substitutes for physical models (Aboelyazeed et al., 2023; Feng et al., 2022, 2023; Höge et al., 2022; Jiang et al.,
2020), or they could use limited physical constraints (Kraft et al., 2022). They are collectively called "differentiable models"
in the sense that they can rapidly compute gradients of outputs with respect to inputs or parameters using automatic
differentiation (or any other means). The differentiability enables the training of neural network components placed anywhere
in the model via backpropagation. Inserting neural networks into process-based models can be perceived as posing questions
regarding some uncertain relationships given some known ones (priors) and we want to get answers for these questions by
automatically learning from big data.

Some of our recent work has applied differentiable modeling to the conceptual hydrologic model named Hydrologiska Byråns
Vattenbalansavdelning (HBV) (Bergström, 1976, 1992; Seibert and Vis, 2012), and built a physics-informed hybrid model for
basins in the contiguous United States (CONUS) (Feng et al., 2022, 2023). The model is "regionalized" in the sense that the
embedded neural network components are trained simultaneously on all basins in the study region in order to provide physical
HBV parameters which are learned from raw information of basin attributes, resulting in improved generalizability and reduced
overfitting to local noise. With the help of differentiable modeling to flexibly evolve the original structure of HBV, the
differentiable hybrid models can approach the performance level of the LSTM model, whilst being constrained to physical
laws and keeping process clarity to predict untrained diagnostic variables with decent accuracy (Feng et al., 2022). Since the
framework is regionalized, this differentiable model can be used to predict in ungauged regions and even extrapolates better
spatially than LSTM in data-sparse regions when tested across the CONUS (Feng et al., 2023).

Owing to the complexity of calibration, current global hydrologic models are largely either uncalibrated (Hattermann et al.,
2017; Zaherpour et al., 2018) or only calibrated on mean annual water budgets or in limited regions (Burek et al., 2020; Müller
Schmied et al., 2014). Only very limited studies attempt to calibrate global models on monthly discharge variations (Werth
and Güntner, 2010). We desire efficient regionalized models that maximally leverage available information and provide
accurate predictions to diverse basins across different climate groups and geographic characteristics in the world. We also want
the models to perform decently even in data-sparse regions, showing competitive extrapolation ability, given that many large
regions such as in Africa and Asia lack publicly available streamflow data. DL and differentiable models seem plausible
candidates for such simulations. Nevertheless, previous studies on DL and physics-informed differentiable models mainly
focus on continental or smaller scales, with a relatively homogeneous forcing dataset --- it is unclear if their observed strengths,
e.g., high performance and strong generalization ability, can carry over to global scales, where the climate is much more diverse
and datasets differ widely in their biases and uncertainty characteristics. In particular, we want to thoroughly examine how
well these models can leverage information learned in data-rich continents to characterize the hydrologic processes in
ungauged regions across the world. Meanwhile, DL models also show favorable scaling relationships (or data synergy) where
more data leads to more robust models (Fang et al., 2022). Thus, training on a larger dataset may provide additional benefits.

In this study, we test physics-informed differentiable models (with the full version name δHBV-globe1.0-hydroDL, where "δ"
represents "differentiable", global1.0 is the version, and "hydroDL" refers to our particular code implementation. δHBV is
used as the abbreviation in this paper) to simulate hydrologic processes for global basins and compare results to purely data
driven methods and traditional modeling approach. We focus on regionalized modeling and emphasize the importance of
spatial generalization in data-sparse scenarios, since observed streamflow data in many parts of the world are scarce. This
means one framework with parameter regionalization from geographic attributes will be used to model all the global basins
rather than calibrating a separate model in each individual basin (Beck et al., 2020b; Feng et al., 2022; Mizukami et al., 2017).
We first investigate what prediction accuracy can be achieved by different models at global scale by learning from a large and
diverse dataset. We then relate the global spatial patterns of model performance to geographic characteristics and hydrologic
processes to identify model structural deficiencies and gain hydrologic insights. Finally, we provide evidence indicating which
type of model may be more appropriate for next-generation global modeling by rigorously examining their generalizability to
ungauged regions across the world.
**2. Data and methods**
**2.1 Global datasets**
We use a global database compiled in a previous study (Beck et al., 2020b) which contains a total of 4229 headwater
catchments. The dataset includes basin mean meteorological forcings, catchment characteristics such as the climate,
topography, land cover, soil composition, and geology information to support parameter regionalization, along with streamflow
gauge discharge observations. Meteorological forcings are the driving inputs of hydrologic models. This global dataset
includes daily precipitation from Multi-Source Weighted-Ensemble Precipitation (MSWEP), a product that merges gauge,
satellite, and reanalysis precipitation data (Beck et al., 2017c, 2019), and maximum and minimum temperature from Multi-
Source Weather (MSWX), a product that bias-corrects and harmonizes meteorological data from atmospheric reanalyses and
weather forecast models (Beck et al., 2022). Potential evapotranspiration was estimated using the method from Hargreaves
(1994). The discharge observations at the outlet gauges were used as prediction targets to train the hydrologic models. We
excluded some basins with potential erroneous discharge records such as showing unreasonable magnitude way larger than
precipitation or dramatic differences between two time intervals, by manually performing visual screening, and also excluded
those with severe amounts of missing data (less than 5 years' worth of data points in the study period from 2000 to 2016).
Thus, 3753 basins were finally used to evaluate different models. These basins had been classified into five Köppen-Geiger
climate classes in Beck et al., (2020b), including tropical (489 basins), arid (109 basins), temperate (1423 basins), cold (1593
basins), and polar (139 basins), as shown in Figure 1. To evaluate the simulations of untrained variables like evapotranspiration
(ET), the MOD16A2GF (Running et al., 2021), a gap-filled 8-day composite ET product estimated from the Moderate
Resolution Imaging Spectroradiometer (MODIS) satellite data and meteorological reanalysis data, were used as independent
observations to compare against the simulated ET from differentiable hydrologic models.

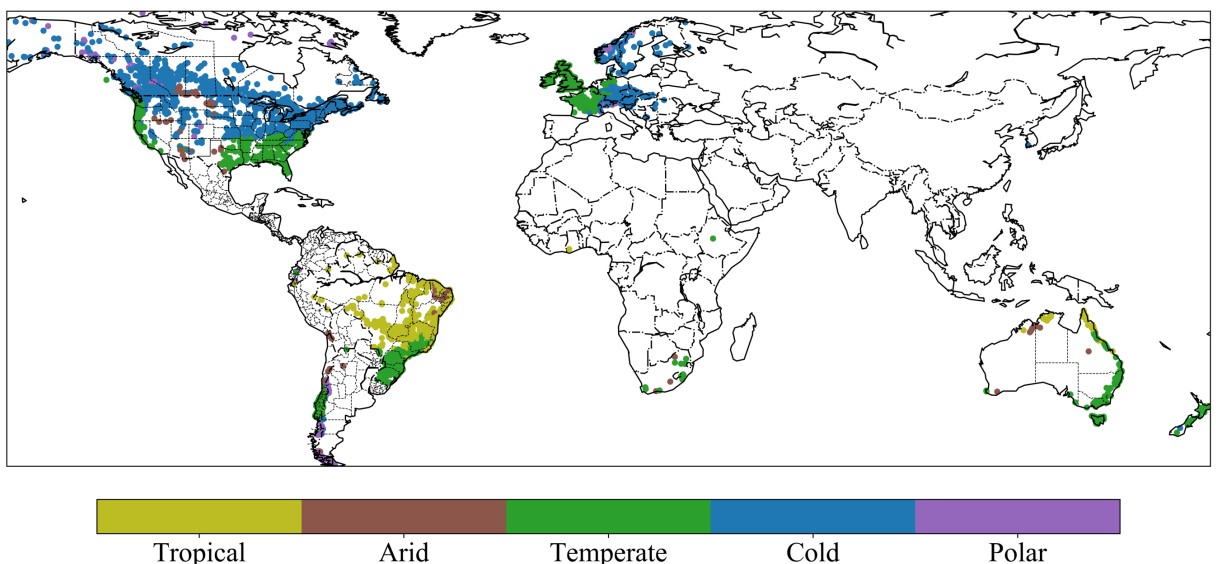

*Figure 1. Locations and climate groups of the 3753 global basins used in this study, which were originally compiled by Beck et al., 2020b.*
*Plotted in Python using Matplotlib Basemap Toolkit.*

## 2.2 The long short-term memory (LSTM) streamflow model for comparison

Here the LSTM model is used as a benchmark for purely data-driven DL. The LSTM has "cell states" and "gates" to maintain
and filter information, as shown in Figure 2a. The input, forget, and output gates control the flow of information, respectively
controlling what to let in, what to forget, and what to output from the system. In this study we use the LSTM streamflow model
demonstrated in Feng et al. (2020) which has been successfully applied to simulate streamflow in hundreds of basins across
the CONUS. The framework takes meteorological forcings and basin attributes as inputs and generates daily streamflow
predictions for each basin at each time step (Figure 2a). We used mini-batches to train the LSTM model, where each minibatch
was composed of two-year sequences from 256 randomly-selected basins. The first-year sequences are only used for
initializing the cell states, so we calculate the batch loss function only on the second-year sequences. The training sequences
were also randomly selected from the whole training period, and one epoch was finished when the model had seen all the
training data. Note that this sequence length is a subset of, and different concept from, the length of training period. Sequence
length specifically refers to the length of the training instance that comprises a minibatch, whereas training period refers to the
whole period when observations are available for training, from which the minibatch sequence length is randomly selected.
The model was forwarded on each minibatch iteratively and its weights were updated using gradient descent after each
forwarding. One epoch was considered to have occurred when the model is iterated over all the training data. We trained the
LSTM model for 300 epochs to achieve convergence.

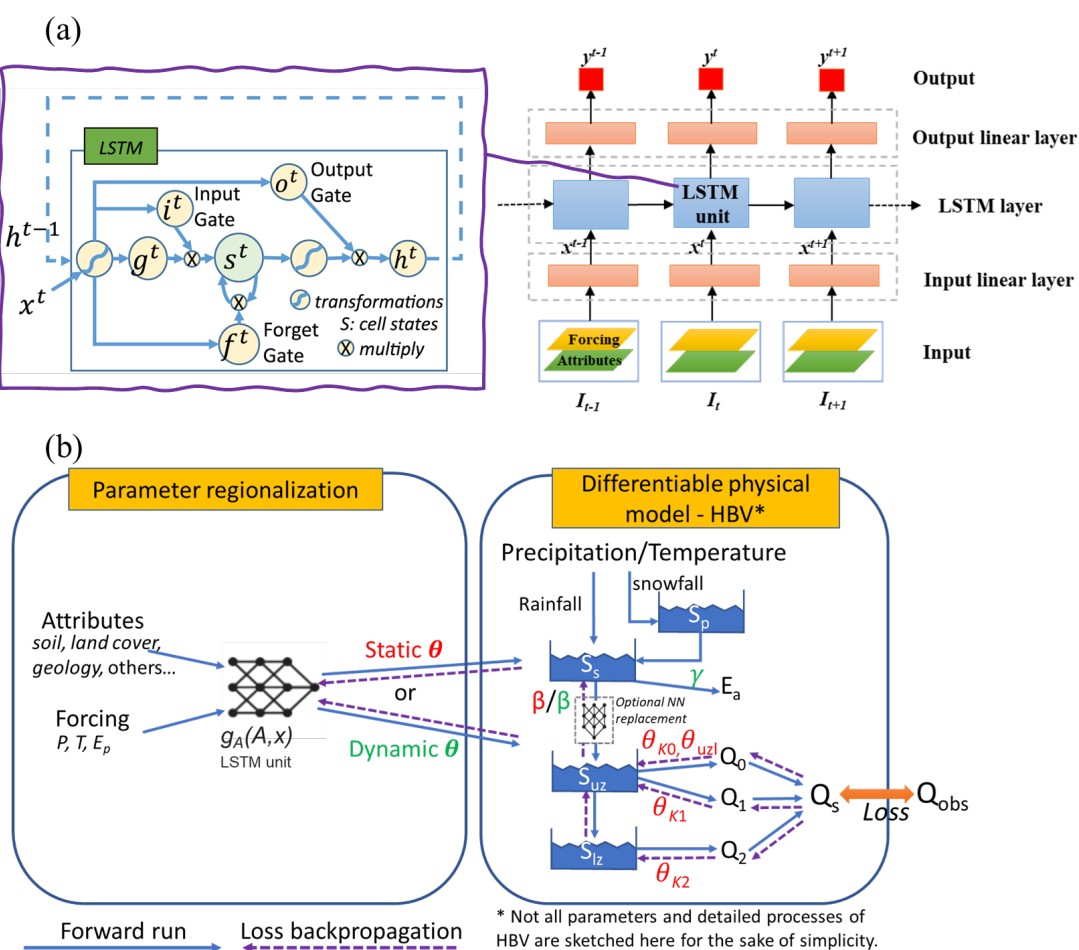


*Figure 2. Illustrations of two different types of regionalized hydrologic models. (a) Framework of the purely data-driven LSTM*
*streamflow model (adapted from Figure 2 in Feng et al., 2020), and (b) framework of the differentiable HBV model (δHBV-globe1.0-*
*hydroDL) with parameter regionalization developed in Feng et al. (2022) (adapted from Figure 1 in Feng et al. (2022)). The neural*
*network $g_A$ here is a LSTM unit which is trained by the observed streamflow to produce the static or dynamic physical HBV parameters*
*$(\theta, \beta, \gamma)$ from basin characteristics.*

## 2.3 The hybrid differentiable hydrologic models

We used the hybrid differentiable models ($\delta$HBV-globe1.0-hydroDL) developed in Feng et al., (2022) for regionalized
modeling in global basins. The HBV model used here as the physical backbone is a conceptual hydrologic model with
representations of snowpack, soil, and groundwater storages, and can simulate flux variables such as snow melting,
evapotranspiration, and quick and slow outflows (Beck et al., 2020b; Bergström, 1976, 1992; Seibert and Vis, 2012). The
differentiable parameter learning (dPL) framework (Tsai et al., 2021) is used to provide parameter regionalization for HBV,
as shown by the $g_A$ neural network in Figure 2b. The $g_A$ network, which is a LSTM unit here, takes basin attributes and
meteorological forcings as inputs, and outputs static or dynamic physical HBV parameters. The differentiable HBV model
then takes these parameters as well as the meteorological forcings to simulate the hydrologic process and predict daily
streamflow discharge along with other key flux variables. The whole framework including HBV itself was implemented in a
DL platform (PyTorch 1.0.1 was used for the original development and the model has also shown good compatibility with
more recent PyTorch versions, (Paszke et al., 2017)) supporting automatic differentiation and trained with gradient descent to
minimize the difference between the simulated and observed streamflow (the loss function). As in Feng et al., (2022), we
employed the loss function based on root-mean-square error (RMSE) with two weighted parts. The first part calculates RMSE
directly on the simulated and observed discharge, while the second part calculates RMSE on the transformed discharge records
to improve low flow representations. Note that we do not directly train the HBV parameters; rather, we focus on training the
weights of the $g_A$ neural network to map the relationship between basin-averaged characteristics and HBV parameters.
Differentiable models are also trained in mini-batches that are formed in the same way as for training the LSTM streamflow
model. Within one epoch, differentiable models are forwarded and optimized over the randomly formed mini-batches until the
iterations have used all the training data points. We train the differentiable models for 50 epochs in total.

As described in Feng et al. (2022), the differentiable modeling framework enables optional modification of the structures of
the original HBV model to enable better performance and we use two versions of evolved HBV models in this study. We used
16 parallel subbasin-scale response units, each with a separate set of parameters to describe a fraction of the basin with different
hydrologic responses. These components implicitly represent subbasin-scale spatial heterogeneity. The simulated fluxes (e.g.,
streamflow) are the average of all the response units. The parameters of the multiple components are different and all are
produced simultaneously by the same $g_A$ network. The first version of our model (referred to as "dPL + evolved HBV") only
has static parameters which are kept constant during the hydrologic simulation. The second version (referred to as "dPL +
evolved HBV with DP) further allows some formerly static parameters of the multi-component model to vary daily with the
meteorological forcings. These dynamic parameters (DP) were also produced by the $g_A$ LSTM unit. If we were to apply the
dynamic parameterization to all parameters, the model could become overly flexible, potentially leading to overfitting to the
training data (which would lead to issues with extrapolation beyond the training data). To reduce the risk of overfitting, we
restricted the dynamism to only two empirical parameters: the shape coefficient $\beta$ in the equation that describes the
relationships between soil storage and potential runoff, and a newly added shape parameter ($\gamma$) which is involved in the
calculation of evapotranspiration. For more details regarding these differentiable HBV models, please refer to our previous
studies (Feng et al., 2022, 2023).
**2.4 Experiments and evaluation metrics**
We ran one temporal and two spatial generalization experiments to evaluate the performance of different regionalized models.
For the temporal generalization experiment, the models were trained for the period of 2000 to 2016 on all global basins, and
tested for the period of 1980 to 1997. Basins without discharge records or with less than 5 years' worth of data points in the
testing period were excluded from the evaluation. Without spatially holding out any basin during training, this experiment
aimed at evaluating the model's generalizability in the time dimension by testing prediction ability on the same basins but in
a different time period from the training data. The other two spatial generalization experiments served as the true litmus tests
for evaluating the effectiveness of regionalization schemes, i.e., how well the model can be applied to basins that have never
been seen during training. The first spatial generalization experiment was a traditional "prediction in ungauged basins" (PUB)
problem, where we randomly divided the whole global basin set into 10 folds (groups) and performed cross-validation across
these folds to obtain spatial out-of-sample predictions for all basins (training on 9 of the folds with the 10th fold held out and
testing on the 10th, then rotating such that each fold is used for testing once). The second spatial generalization experiment,
which we refer to as cross-continent "prediction in ungauged regions" (PUR), was more challenging. In this experiment, we
assumed that all the basins in certain continents are ungauged and excluded from the training dataset, trained a regionalized
model in other data-rich continents, and then tested the trained model to make predictions in the ungauged continents. With
random hold-out, an ungauged test basin in the first spatial generalization experiment always has training gauges surrounding
it. Therefore, the first PUB experiment can be interpreted as spatial interpolation. The second spatial experiment (cross-
continent PUR) holds out all the basins in one continent as testing targets, and thus is the much harder test of spatial
extrapolation.

To evaluate the overall performance of the hydrologic models, we used the Kling-Gupta Efficiency (KGE) (Gupta et al., 2009;
Kling et al., 2012) as compared in Beck et al., (2020b) and Nash-Sutcliffe Efficiency (NSE) (Nash and Sutcliffe, 1970). KGE
has three components that account for correlation, mean bias (the ratio of simulated and observed means), and variability bias
(the ratio of simulated and observed coefficients of variation), while NSE mainly represents the variance explained by the
simulations. Both metrics indicate better performance when their values are closer to the maximum value of 1. We also
examined the percent bias of the top 2% peak flow range (FHV) and bottom 30% low flow range (FLV) of streamflow
predictions to evaluate the model's ability to simulate extreme events (Yilmaz et al., 2008). All the reported performance
metrics in this study are from model evaluation on the testing dataset, which is not seen by the model during the training
process.

## 3. Results and discussions

### 3.1 General patterns over global basins

From the standpoint of daily hydrograph metrics (KGE and NSE), LSTM and the two differentiable models all achieved highly competitive performance for the global basins in the temporal test (trained and tested on the same basins, but in different time periods) (Figure 3). For the global dataset, all three models obtained median KGE values close to or higher than 0.7, but the LSTM model performed the best of the three models here, achieving a median NSE (KGE) value of 0.70 (0.74) for all the evaluated basins. For a subset of 1675 basins with long-term records (at least 15 years' worth of streamflow data available in the training period and 5 years' worth of data available in the testing period, though not necessarily continuous), LSTM even reached a median KGE of 0.78 (see Figure A1). Both versions of the differentiable models approached the performance level of the LSTM, in agreement with our previous assessment for the CONUS (Feng et al., 2022). The model with dynamic parameters achieved a median NSE (KGE) of 0.67 (0.69), followed by the model with static parameters, which obtained a median NSE (KGE) of 0.65 (0.68).

The LSTM exhibited advantages for the low flow predictions compared with the differentiable models, as shown by the FLV metric (Figure 3). However, for the peak flow predictions, the LSTM and differentiable models were quite similar, and they all underestimated the observed peaks (FHV in Figure 3). The underestimation for peak flows is consistent with what was found in previous studies. For example, all the physical and deep learning models have significant negative peak flow bias when benchmarked in the CONUS dataset (Feng et al., 2020; Kratzert et al., 2019b). We hypothesize that the systematic underestimation of peaks may be partially related to bias in precipitation forcings. MSWEP is based on the ERA5 reanalysis, which is known to underestimate precipitation peaks (Beck et al., 2019). Furthermore, the use of basin-averaged, daily-averaged precipitation may further suppress the peaks (Chen et al., 2017). In addition, the errors with peak flow could also be partly due to some numerical and structural issues with the differentiable models, e.g., numerical errors introduced by the explicit and sequential solution scheme of HBV with excessive use of threshold functions that lead to different results when the sequence changes, and structure limitations, e.g., deeper groundwater storage cannot feed back to the upper layers. Given the commonality of this issue, we call for community efforts and collaboration to address this issue.

Both the LSTM model and the differentiable models performed well over diverse landscapes, including North America (especially along the Rocky and Appalachian Mountain ranges and the Southeastern Coastal Plains), Western Europe, Asia (mostly Japan), the southern part of Brazil, and the northeast coast of Australia (Figure 4a and b). There are other regions where none of the three models performed well, such as the longitudinally-central part of North America (Great Plains and Interior Lowlands), the southern edge of Chile (with many glaciers), the Tasmania state of Australia, and the few basins in Africa. These regions, for example, the Northern Great Plains and the state of Texas in the CONUS, have always been difficult for all kinds of models, likely due to incorrect basin boundary, highly localized precipitation, the dry conditions with small

runoff amounts and flash flooding mechanisms (Berghuijs et al., 2014; Driscoll et al., 2002; Feng et al., 2020; Martinez and
Gupta, 2010; Newman et al., 2017), to be explored below. Despite some challenges, however, these values represent currently
the best metrics reported at the global scale compared to earlier studies, e.g., (Alfieri et al., 2020; Beck et al., 2017a, 2020b;
Hou et al., 2023), attesting to these models' great potential as global modeling tools.

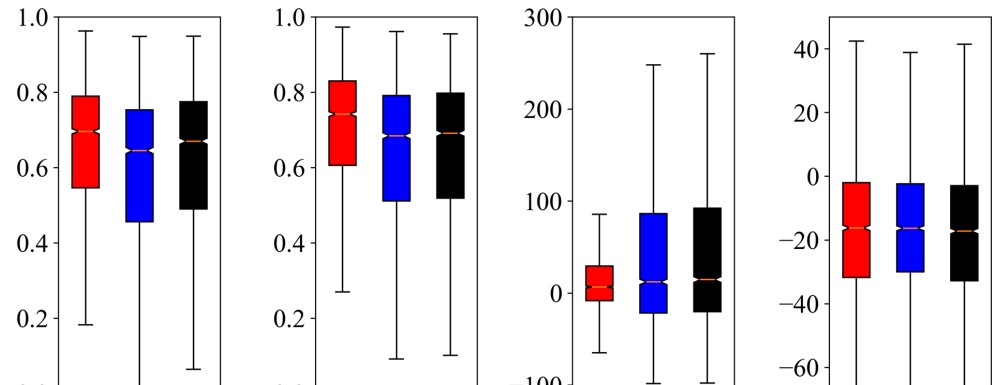


*Figure 3. Performance comparison between the LSTM and differentiable models on global basins. dPL refers to the differentiable*
*parameter learning framework, while "evolved HBV" refers to some modifications to improve the standard HBV model, and "with DP"*
*indicates that some parameters were allowed to be dynamic rather than static. Here, the horizontal line inside the colored box represents*
*the median, while the top and bottom of the colored box indicate the first and third quartiles. The bars extending from the colored boxes*
*indicate 1.5 times the interquartile range from the first and third quantiles. NSE is Nash-Sutcliffe Efficiency, KGE is Kling-Gupta*
*Efficiency, FLV indicates the model's percent bias on the bottom 30% low flow range of streamflow, and FHV indicates percent bias on*
*the top 2% peak flow range of streamflow.*

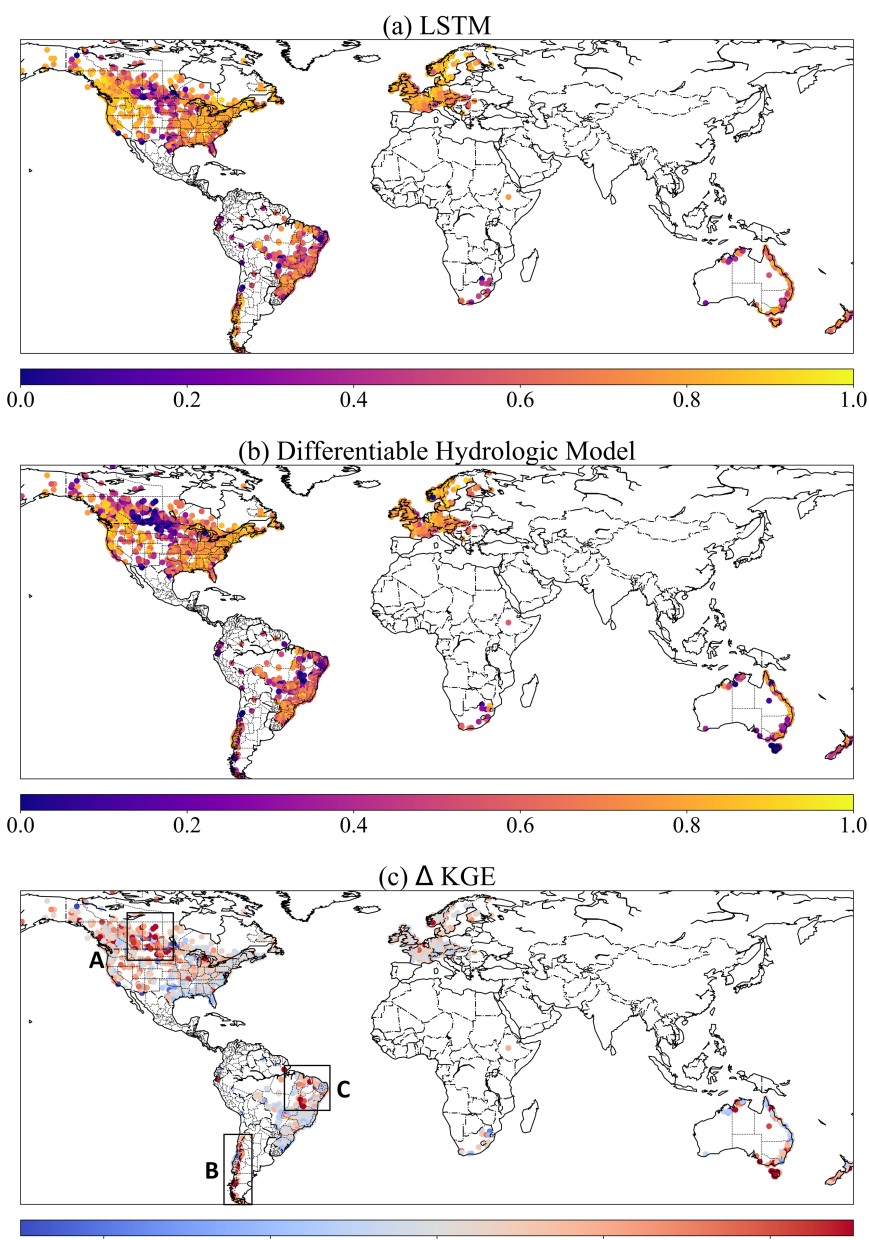


**Figure 4. The spatial patterns of different model performance and their differences shown by KGE metric. (a) the LSTM model; (b) the differentiable model with dynamic parameters (dPL + evolved HBV with DP); and (c) the KGE difference between two models (KGE of LSTM – KGE of dPL + evolved HBV with DP). Plotted in Python using Matplotlib Basemap Toolkit.**

## 3.2 Model behaviors and limitations across climate groups and regions

All three models' performances vary significantly across different climate groups of the global basins (Figure 5), revealing their strengths and limitations. The LSTM model behaved the best in the polar, cold, and temperate groups, while the performance deteriorated in the tropical and arid basins. Similar to LSTM, differentiable models showed strong performance in temperate and cold groups and worse performance in tropical ones, with the worst performance in arid basins. These clusters of challenging basins can also be identified on the map (Figure 4a and b). The differentiable model with dynamic parameters performed better than the model with static parameters in all climate groups except the most challenging arid group. Dynamic parameterization with more structural flexibility generally provides stronger modeling ability, while also showing a higher risk of overfitting and degraded generalizability in basins which are very difficult to simulate. As we examine how LSTM and differentiable models behave differently, we find that such differences can be attributed to processes missing from the simple backbone process-based model (HBV here) as explained below. Here we use LSTM as an indicator of upper bound, that is, it shows the ideal performance of a model, given the available information from forcing and input data. Thus the distance from LSTM indicates either systematic and predictable forcing errors (which can be remediated by LSTM) or structural issues with the differentiable model.

For example, the polar group stands out as a climate type favoring LSTM, while the cold group shows a similar but less pronounced contrast, both of which may be related to HBV's physical deficiencies and forcing issues with snow undercatch. For the polar (cold) groups, LSTM surprisingly had a median KGE of 0.81 (0.78) while the differentiable model only reached 0.62 (0.71). The polar regions include, for example, Southern Chile (in region B in Figure 4c). As glaciers can store water for extended periods of time and are driven mostly by temperature rather than rainfall, it is possible for LSTM to capture the temperature-driven dynamics (Lees et al., 2022) while the original HBV itself does not have a glacial module. HBV does not have the ability to simulate frozen soil, sublimation or snow cover fractions. Furthermore, as snow gauges in high altitude are known to suffer systematic bias due to undercatch problems (Beck et al., 2020a), LSTM can learn to address such systematic bias while physical differentiable models cannot due to mass balance. For the cold regions, e.g., high-latitude regions of the North American Great Plains (Region A in Figure 4c --- this also includes the Prairie Pothole Region, or PPR), HBV may suffer from not having descriptions for frozen ground conditions (soil ice) which can influence infiltration, and rainfall underestimation due to undercatch, ice blockage, and other potential reasons (Beck et al., 2020a). In addition, another reason why LSTM and differentiable HBV may have trouble with PPR (but HBV performed especially poorly) is the countless wetlands that store water until full and become connected after snowmelt and large rainfall. HBV does not have modules that can describe such large-scale fill-connect-spill processes (Shaw et al., 2013; Vanderhoof et al., 2017).

A more prominent challenge is the arid regions (middle CONUS, north Chile and east Brazil in Figure 1 and Figure 4). This challenge can be attributed to the long duration of low flows which requires long-term memory, and flash floods which result

from intense short-duration storms not well represented at the daily scale. Even the LSTM model cannot retain year-long
memory and cannot perform well for the baseflow (Feng et al., 2020). Because HBV has a linear reservoir for its slow-flow
(lowest) bucket, it cannot generate zero base flows. Neither can it well simulate the impact of intense hourly-scale rainfall.
These process improvements need to be considered in the future. Another reason for the challenge in arid regions is the lack
of reservoir management modules. Arid regions tend to have water management infrastructure that significantly influences
streamflow (Veldkamp et al., 2018). Since the HBV model doesn't have any module representing human impacts on the natural
water cycle, the poor performance in middle Brazil in region C may have come from the missing representation of human
interferences. There are large population and intensive agricultural activities in this region which could induce significant
impacts on the hydrologic process. Parameter compensations apparently cannot make up for all the missing mechanisms.

The sensitivity of model performance to missing processes in the differentiable models is both good and bad news. It's good
news because this means we can identify suitable or insufficient process representations by learning from data. On the other
hand, this means more challenges as we need to increase the process complexity of this model before it can perform well for
these basins, unlike the purely data-driven LSTM which is not explicitly concerned with physical processes.

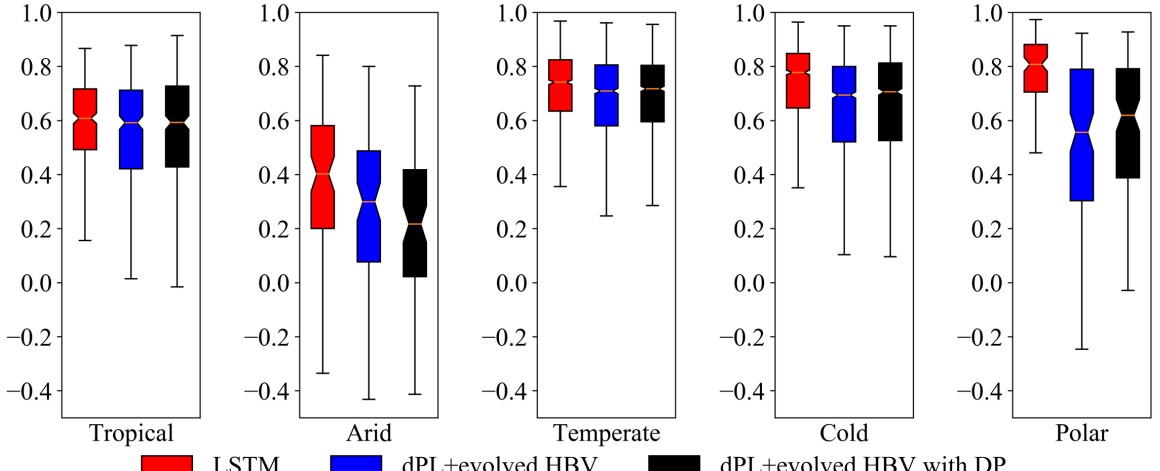


*Figure 5. The performance comparison (KGE, Kling-Gupta Efficiency) of different models for five climate groups. dPL refers to the*
*overall differentiable parameter learning framework, while "evolved HBV" refers to some modifications to improve the standard HBV*
*model, and "with DP" indicates that some parameters were allowed to be dynamic rather than static. Here, the horizontal line inside the*
*colored box represents the median, while the top and bottom of the colored box indicate the first and third quartiles. The bars extending*
*from the colored boxes indicate 1.5 times the interquartile range from the first and third quantiles.*
**3.3 Spatial generalization for prediction in ungauged regions**
While LSTM maintains mild advantages over differentiable models in data-dense settings, it was outperformed by
differentiable models in a highly data-scarce scenario. As mentioned above, the data-dense setting was tested in the randomized
holdout test called prediction in ungauged basins (PUB), while the data-scarce scenario was tested in the regional holdout test,
or prediction in ungauged regions (PUR). In the global PUB test, LSTM has a small edge (median KGE=0.67) over
differentiable models (median KGE=0.64). Both were noticeably higher than the traditional regionalization method using
linear transfer functions reported by Beck et al. (2020b) (Beck20, median KGE=0.46), which already represents the previous
state-of-the-art performance of global parameter regionalization. Differentiable modeling does not rely on strong assumptions
of the functional form for the parameter transfer function. It leverages the powerful ability of neural networks to represent
complicated functions, and automatically learns robust and generalizable relationships between geographic attributes and
physical model parameters from large data. Therefore, we can expect significant performance advantages from differentiable
modeling compared to traditional methods relying on linear transfer functions. In the PUR scenario where European basins
were held out for testing, differentiable models (median KGE=0.58) performed significantly better (p-value less than 0.01
using the one-sided Wilcoxon signed-rank test) than LSTM (median KGE=0.52). In the South American PUR experiment,
lower performance was seen for all models which can be expected considering the prediction difficulties in this region even
for the in-sample scenario (Region B and C in Figure 4). The median KGE of LSTM is 0.28 while the differentiable model
with static parameters achieves a higher median KGE of 0.31 for the PUR scenario. It seemed that the differentiable model
with dynamic parameterization was somewhat overfitted in this case, resulting in a median KGE that was lower than the static-
parameter differentiable model. We do not have PUR results from traditional models available to compare against, since this
is a very challenging issue for traditional regionalization methods to make predictions across continents.

With these results, we show that differentiable models have demonstrated a high simulation capability that cannot be obtained
with traditional parameter regionalization approaches, and also provide a robust extrapolation capability in large data-sparse
regions that is stronger than purely data-driven models like LSTM. This conclusion was not only verified in the USA, but now
has also been confirmed in global catchments with generalization tests including prediction in neighboring ungauged basins
and cross-continent predictions, each of which have different conditions with respect to data availability and density.

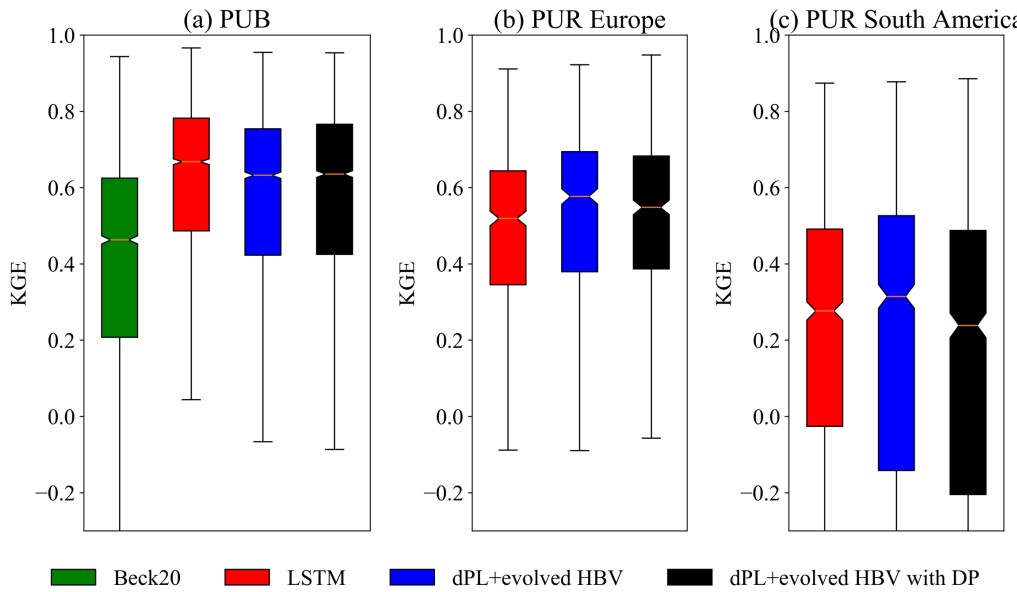


*Figure 6. The performance comparison (KGE, Kling-Gupta Efficiency) of different models for spatial generalization tests. (a) Random*
*hold-out test for prediction in ungauged basins (PUB), (b) and (c) holding out all the basins in Europe or South America, respectively,*
*for cross-continent predictions in ungauged regions (PUR). Beck20 refers to a traditional regionalization method using linear transfer*
*functions (Beck et al., 2020b), LSTM is the purely data-driven long short-term memory network, dPL refers to the differentiable*
*parameter learning framework, while "evolved HBV" refers to some modifications to improve the standard HBV model, and "with DP"*
*indicates that some parameters were allowed to be dynamic rather than static. Here, the horizontal line inside the colored box represents*
*the median, while the top and bottom of the colored box indicate the first and third quartiles. The bars extending from the colored boxes*
*indicate 1.5 times the interquartile range from the first and third quantiles.*

**3.4 Predicting untrained variables**
The evapotranspiration (ET) simulations from differentiable models are consistent with independent MODIS satellite estimates
of ET in both temporal dynamics and spatial patterns. We did not use any ET observations as training targets to supervise the
differentiable models. At the global scale, the mean annual ET comparison shows overall consistency with MODIS, with most
basins lying close to the 1:1 line and a correlation of 0.75 for all the basins (Figure 7a). Spatially, the model was able to
represent energy limitations in the cold regions, e.g., high-latitude North America and Europe, and water limitations, e.g.,
southwestern US and arid basins of Australia (Figure 7a and b). The model also represented high ET in basins adjacent to the
Amazon forest, those along the US southeastern and Australian coast. Temporally, the median correlation of ET time series
between simulations and MODIS products achieves 0.82 and 0.89 for two differentiable models in 3753 basins, respectively
(Figure 7c).

The ET simulations show high correlation with MODIS in most North American and European basins (Figure 7d), in line with
the good performance of streamflow modeling in these regions. However, the correlation is relatively lower in South America
but the coefficient of variation of ET residuals (CoV, the ratio of standard deviation of ET residuals to the annual mean) is also
small (Figure 7e), in part because the ET here is large and less driven by the seasonal energy cycle (Niu et al., 2017). MODIS
ET itself is not the ground truth and always has large uncertainties in Amazonia regions due to the cloud coverage and
difficulties for observation (Hilker et al., 2015; Xu et al., 2019). Furthermore, the simulations could be negatively influenced
by the data quality issues with streamflow records in these regions. Upon examining the records, some stations in South
America show unrealistic hydrographs that may indicate data processing errors. To address such issues in the future, more in-
depth data screening and correction or constraining the model using datasets other than streamflow, e.g., eddy covariance flux
data, should be considered. The CoV is less than 0.3 for most of the world, showing that ET errors are mostly small relative to
its annual averages (Figure 7e). Noticeable exceptions are US southwest, where ET varies strongly from year to year and is
highly dependent on the precipitation, and Chile, where glaciers and deserts are both present, posing challenges to the model.
As the present study is basin-focused, we will leave the evaluation of global gridded ET to future work.

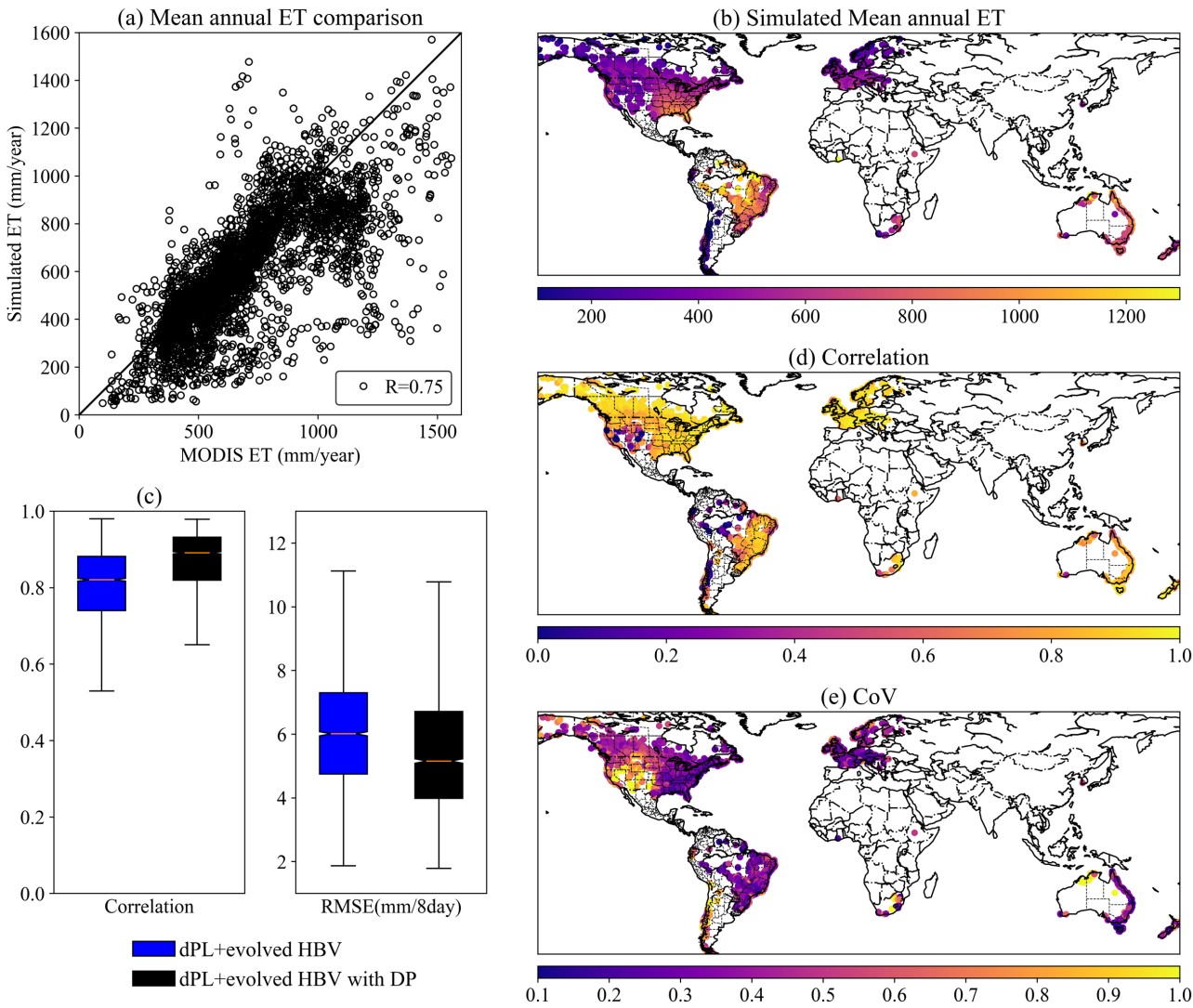

*Figure 7. The comparison between simulated ET from the differentiable hydrologic models and independent MODIS ET product. (a) mean annual ET comparison, (b) simulated mean annual ET for global basins, (c) boxplots for the temporal dynamic evaluation by correlation and RMSE, (d) correlation and (e) coefficient of variation for ET comparison in global basins. Maps plotted in Python using Matplotlib Basemap Toolkit.*

**3.5 Further discussion**

Compared to the LSTM model which only outputs discharge simulations, differentiable models offer a suite of interpretable variables including ET, soil water, recharge, baseflow, etc., thus providing a comprehensive description for the hydrologic cycle and far better interpretability. To create a new differentiable model or turn an existing model into a differentiable one, we need to implement the model on a differentiable platform like PyTorch, Tensorflow, or JAX, while better enabling model parallelism in order to maximally leverage the computing power of modern graphical processing units (GPUs). If a model

contains mostly explicit calculations, automatic differentiation (AD) offered by the above platforms can effortlessly provide
gradient calculations, requiring only a syntax-level translation which can nowadays be done easily. Sometimes, a limited
amount of adjustments are needed to turn non-differentiable operations into equivalent differentiable ones. However, when a
model contains iterative solutions to nonlinear systems, large matrix solvers or constrained optimizations, we can employ the
adjoint method (Song et al., 2023). The adjoint method explicitly defines the gradient-calculation method and alters the order
of calculations so iteration is avoided during gradient calculations, which can dramatically reduce memory demand and
improve efficiency. Another important consideration is the effective use of parallelism and the modern computing
infrastructure for AI (i.e., GPUs). In our context, the regionalized parameterization (in this case, training one neural network
on a large amount of basins), which is crucial to ensuring the generalizability of the model, requires going through large data
in high-throughput parallelism. Embracing parallelism may necessitate some coding adjustments. At this point, several
versions of differentiable hydrologic models have been proposed with varying complexities and different handling of
parameterization, post-processing (which we didn't use in this study, as it can interfere with interpretability of the internal
variables, mass balances, and the sensitivity to inputs encoded by the process-based components), and dynamical parameters.
Across geoscientific domains, differentiable ecosystem (Aboelyazeed et al., 2023; Zhao et al., 2019), flow and routing (Bindas
et al., 2024), water quality (Rahmani et al., 2023), and ice sheet (Bolibar et al., 2023) models have already been demonstrated.
The challenges facing the differentiable models in this study include not only missing processes like reservoir management,
ground ice, and glaciers, but also large errors in meteorological forcings and streamflow target data. Substantial bias could
exist in precipitation, e.g., due to snow-gauge undercatch (Hou et al., 2023), or in discharge, e.g., streamflow are measured
using different approaches which exhibit large variability; for another example, gridded climate forcing data often consistently
underestimate the magnitudes of heavy storms (Beck et al., 2017b). While LSTM can easily adapt to systematic bias, such
forcing errors put the differentiable models under stress because they cannot reconcile streamflow observations with such
forcings given the constraint of mass balances. If our objective is to learn core physics and parameterizations that are reliable
despite forcing discrepancies, we can set up forcing data correction layers that can, to some extent, shield the core processes
from being influenced by such errors. This will be an important aspect of future work to ensure reliable prediction of future
water resources.
The backbone of a differentiable process-based model thus serves as a double-edged sword: when such backbones are
essentially correct, they serve as a stabilizing element of the model that mitigates overfitting and improves generalization;
when they lack critical processes or when observations have large, unexplained bias, they can drag down model performance
and cause compensation between processes. However, the limitations are tractable: future work can gradually incorporate
critical processes and include more observations to constrain the learning process, making sure each addition is valuable and
accretive. The research community collectively has already substantial experience in evolving earth system models to include
many processes. We expect some processes to be invited back in the differentiable modeling framework. Nevertheless, with
differentiable modeling, we now have a new tool that was not previously available: highly flexible deep neural networks that
can be placed anywhere in the model, which provide a systematic way of managing model complexity. With their help, such
model evolution may take much less time than previously required. However, we still expect the development cycle to take
longer than that for purely data-driven models like LSTM, requiring us to view differentiable models as evolving rather than
static entities, which need a bit of patience while maturing.

This study builds a benchmark and a basis for model selection and diagnosis for the next-generation global hydrologic
modeling, which previously did not learn from such large observations. With rigorous tests at the global scale, this study proves
that differentiable models are strong candidates as global water models. With powerful spatial generalization ability, they can
be applied to characterizing the hydrologic processes in ungauged regions by leveraging learned information in data-rich
continents. Differentiable models in this study have already learned the generalizable and robust relationships between
geographic attributes and physical model parameters from thousands of global catchments. Therefore, these models can be
easily applied towards providing seamless global hydrologic modeling with parameters directly generated from worldwide
geographic attributes. Future work can use such models and continuously improved observational datasets to produce global
hydrologic fluxes while enhancing some process representations in extremely arid, glaciated, or heavily human-influenced
basins.
**4. Conclusions**
In this work, we used both purely data-driven models and, for the first time, physics-informed, differentiable models to simulate
rainfall-runoff processes in 3753 global basins. Both types of models achieved overall highly competitive performance for
global basins with diverse climate conditions, yielding median KGE values close to or higher than 0.7 which is state-of-the-
art at this large scale. The LSTM still achieved the best performance for the temporal generalization test, but the differentiable
HBV models with evolved structure (δHBV-globe1.0-hydroDL) approach the LSTM's performance level. Furthermore, the
spatial generalization experiments highlighted the stronger regionalization and extrapolation ability of differentiable models
than the traditional modeling approach and LSTM, demonstrating its promise to be applied to data-scarce regions in the world.
River routing is not included in this work and will be investigated in the future, possibly also with differentiable approaches
(Bindas et al., 2022).

Different models appear to have generally consistent spatial performance patterns, though obvious distinctions stand out in
several local regions. All models achieve good performance in the temperate and cold climate groups, while they all behave
unsatisfactorily in the arid group. For the polar group, the differentiable model performed significantly worse than the LSTM.
Without any physical constraints, LSTM shows strong power in simulating storage (snow and glacier) dominated processes,
while differentiable models are limited by the structure of their physical backbone model, which in this case does not simulate
multiyear ice buildup and melt. Another limitation could be soil sealing processes in extremely arid regions. These regional
performance comparisons thus reveal some deficiencies of the physical backbone in δHBV that cannot be mitigated even by
advanced neural network-based parameterization. These insights provide directions for future improvements. Different from
purely data-driven models only trained by the target variable, differentiable models constrained by the physical backbone can
give accurate simulations for a full set of hydrologic variables in the water cycle including evapotranspiration, snow water
equivalent, water storage, infiltration and baseflow. As some process limitations are addressed in the future, we believe
differentiable models will be strong candidates for the next generation global water models to characterize and predict the
hydrologic processes in ungauged regions across the world.
**Appendix**

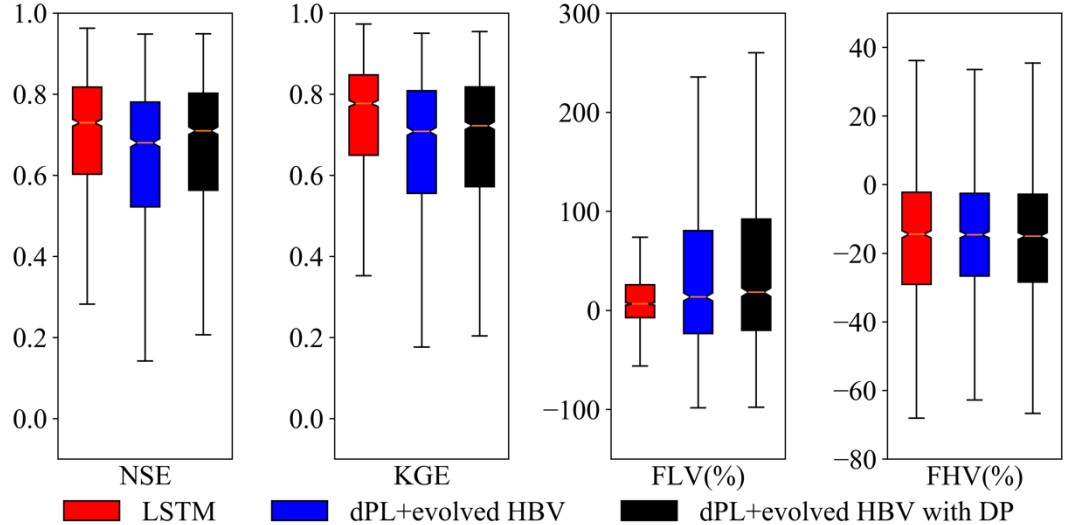

*Figure A1. Performance comparison on the 1675 subset basins with long-term streamflow records (at least 15 years' worth*
*of streamflow data available in the training period and 5 years' worth of data available in the testing period, not necessarily*
*continuous). Other items are the same as in Figure 3.*

**Author contributions**
DF and CS conceived this study. DF set up the hydrologic models and ran all the experiments. DF and CS performed the major
analysis, with HB, JdB, RKS, YS, YW and MP contributing substantially to the discussions on the methodology and results.
HB provided the benchmark results of the traditional parameter regionalization scheme. JL prepared the ET product for
comparison. DF wrote the initial draft and CS revised the manuscript. HB, JdB, RKS, YS, YW, and KL substantially edited
the manuscript.

**Financial support**

DF was supported by the National Science Foundation Award EAR-2221880. This work was also partially supported and inspired by the Young Scientists Summer Program (YSSP) of International Institute for Applied Systems Analysis (IIASA). JL was supported by Google.org's AI Impacts Challenge Grant 1904-57775. CS and KL were supported by Cooperative Institute for Research to Operations in Hydrology (CIROH), award number A22-0307-S003. Computation was partially supported by the National Science Foundation Major Research Instrumentation Award PHY-2018280.

**Competing interests**

Kathryn Lawson and Chaopeng Shen have financial interests in HydroSapient, Inc., a company which could potentially benefit from the results of this research. This interest has been reviewed by The Pennsylvania State University in accordance with its individual conflict of interest policy, for the purpose of maintaining the objectivity and the integrity of research at The Pennsylvania State University.

**Code and Data Availability**

The source codes for the differentiable hydrologic models can be accessed at https://doi.org/10.5281/zenodo.7091334, and this study evaluates these models at global scale. The MOD16A2GF ET product can be downloaded at https://lpdaac.usgs.gov/products/mod16a2gfv061/. Meteorological forcing datasets MSWEP and MSWX can be downloaded at https://www.gloh2o.org/mswep/ and https://www.gloh2o.org/mswx/, respectively. The streamflow observations used in this study were initially compiled by Beck et al., (2020b) and can be accessed from the original data sources including the United States Geological Survey (USGS) National Water Information System (NWIS; https://waterdata.usgs.gov/nwis), the Global Runoff Data Centre (GRDC; https://grdc.bafg.de), the HidroWeb portal of the Brazilian Agência Nacional de Águas (https://www.snirh.gov.br/hidroweb), the European Water Archive (EWA) of EURO-FRIEND-Water (https://www.bafg.de/GRDC/EN/04_spcldtbss/42_EWA/ewa.html) and the CCM2-JRC CCM River and Catchment Database (https://data.jrc.ec.europa.eu/collection/ccm), Water Survey of Canada (WSC) National Water Data Archive (HYDAT; https://wateroffice.ec.gc.ca/), the Australian Bureau of Meteorology (BoM; http://www.bom.gov.au/waterdata/), and the Chilean Center for Climate and Resilience Research (CR2) website (https://www.cr2.cl/datos-de-caudales/).

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
