# Peer review of "Deep Dive into Hydrologic Simulations at Global Scale: Harnessing the Power of Deep"

_Geoscientific Model Development, 2023_

## Author Comment (AC2)

This study presents a simulation using a differentiable model at 3753 basins globally. It represents an advance in the field and is thus worthy of being published somewhere. However, the used model and the design of the experiments are almost the same as those published in the authors' previous papers. Readers of GMD would expect more progress in those aspects. Thus, I suggest a major revision.

Dear reviewer,

We thank you for your comments. We would like to note, although the model has not been changed, the dataset and training changed drastically. It expanded from data from only the USA to the whole globe, and this is no easy feat. Enormous effort went into cleaning the data, improving model efficiency, training and testing, and interpreting the results. It was highly uncertain if the model qualities would carry to such a large dataset. Many of the insights are new. Plus, we do want to ensure some consistency between the evaluation across datasets so we understand how the same model behaves on different datasets.

We will revise the introduction to better highlight the questions.

GMD does have model evaluation paper type and there are many papers on GMD which evaluate the same model under different scenarios. In fact, we can probably say that it is a minority case to be changing the model structure significantly in each paper. That is why we think the paper fits GMD and will be a valuable contribution.

Major comments

- The authors should introduce more details of the experimental design, such as the metrics used in the training, how many experiments (temporal generalization, PUB, and PUR; correct me if I am wrong), and the purpose of the experiments. Some details may have been presented somewhere else. I find this manuscript difficult to follow without reading the authors' previous publications.
Thanks for the comment. We will revise the manuscript to give some more background on earlier work, and add details like metrics, experimental design, etc.

- L124: what are the criteria for the selection? Can you describe the erroneous cases? Do the erroneous cases include the data processing error described in L350?

There are some data records that just seem totally impossible, for example, total discharge that are way larger than precipitation, yet unable to be linked to some simple unit conversation errors. We will clarify in the revised manuscript that, yes, L124 and L350 refer to the same group of gages.

- L306-L317: Can you discuss more about comparing the traditional regionalization method and PUR? The PUR regionalization method can utilize a large number of observations to

calibrate/train the differentiable model, whereas the traditional method can only use very limited samples. In other words, PUR may have a much higher chance of finding the optimal parameters than the traditional method.

We will add to the discussion to clarify this. It has been established in the past work that differentiable has a better performance than MPR (Tsai et al., 2021) and another regionalization scheme. MPR actually has a similar concept as dPL, but it only uses linear functions because the model is not differentiable and they cannot optimize a large number of weights (as in neural networks) like the differentiable models can.

[Figure]

**Fig. 6 Comparison of dPL and regionalization schemes for streamflow calibration. a** Calibrating the VIC hydrologic model via a differentiable surrogate model on the CAMELS dataset over the conterminous United States, in comparison to Multiscale Parameter Regionalization (MPR). **b** Calibrating the HBV hydrologic model (not a surrogate) on the Beck20 global dataset, in comparison to the Beck20 regionalization scheme. We used NSE for the CAMELS case and KGE for the global case because these metrics were used by the respective papers, and the main purpose of the case studies was to compare with the existing literature. For both panels, curves on the right represent better models.

The above figure is from Tsai et al., 2021. Compare that with our model today:

[Figure]

(green line is the model benchmarked in this paper). You can see that we have made significant progress, so while we cannot run MPR directly, it would be logical to derive that the current version of the differentiable HBV is a lot stronger than MPR.

The biggest power of differentiable model is that it can learn robust relationships from big data. Hence what the reviewer said is true, but we would modify it as "*differentiable models may have a much higher chance of finding the most high-performing, robust and transferable parameters than the traditional method.*"

Minor comments

- Title: the phrase, global hydrologic simulations, is misleading. The simulations are conducted at 3753 basins across the globe. It represents a concept different from the "global hydrologic simulations."
Will revised to "Deep dive into hydrologic simulations at global scale"

- L127: is the classification from Beck et al., 2020b?
We will clarify this.

- L26 & L212: How is the subset of 1675 basins selected? What is the objective of the selection?
The selection criteria was long-term data availability. We will clarify this.

- L223: can you describe more about the structural issues? Why does the explicit solution scheme introduce numerical errors here?
Here we mentioned numerical error introduced by the explicit scheme. There are also issues related to how groundwater storage cannot feedback to the upper layers, which will be mentioned upon revision. However, we do not know all the issues (otherwise we would have fixed them).

- L291-L293: please rewrite the sentence. It is difficult to read.
Will revise to "*The sensitivity of model performance to missing processes is both good and bad news. It is good news because this means we can identify better processes from data...*"

- L398, "the underrepresentation of the processes...": this conclusion is too general. The difficulty of representing arid/polar/anthropogenic processes is known before reading this paper. The conclusion should be specific.

Well. There are many structural deficiencies with HBV, but it seems the parameterization approach can make up for some of the deficiencies. Rather here we are saying these two cases even the NN-based parameterization cannot make up for these missing processes. We will clarify upon revision.

---

## Author Comment (AC3)

R2

Overall, the manuscript is well-written with a clear research objective, innovative hybrid models, and solid results. The study compared the performance of the commonly used LSTM model with two differentiable hydrological models (static and dynamic parameters) with a temporal generalization experiment and conducted a comparative analysis for traditional "prediction in ungauged basins" problem. The model evaluation results provide valuable insights into improving the mechanism of the hydrologic models, confirming the strong localization and extrapolation capabilities of differentiable models. Below are some key comments and concerns:

Thank you for your evaluation.

L161-166: From my understanding, the original parameter calibration process of the HBV model has been replaced by the parameter calibration process of the $g_A$ neural network. If this is the case, it is still necessary to run the HBV model. How does this approach compare to the traditional hydrological model calibration methods in terms of modeling speed? Has it resulted in time and labor savings in the modeling process?

Good question. It resulted in orders of magnitude of computational savings, and we'd argue labor saving as well, but it depends on the economy of scale. Below we will provide some background. We realize that many people have not read our previous paper (Tsai et al., 2021), and we will include some of discussion below in the revised manuscript.

There is an obvious time saving in parallelism. During training, we run models in parallel over a "minibatch", which means somewhere ~100 basins at a time, before we calculate the loss and update the parameters. This is happening all in parallel due to easy GPU concurrency via PyTorch.

There is a less obvious economy of scale, which is that the training and the learned knowledge of neural networks are shared by all sites. This means the more sites participating in the training, the more efficient it becomes (see the figures below, from Tsai et al., 2021). Hence, if you run differentiable model training for one of a few sites, you won't be as efficient as traditional optimization. If you run it for 10000s of sites, you will become orders of magnitudes more efficient. The number we cite in Tsai et al., 2021 is that, for the same calibration job over entire USA, a traditional evolutionary algorithm would need a 100-processor cluster to run 2-3 days, whereas our differentiable parameter learning needs 1 single GPU for 1 hour.

[Figure]

We argue it actually saves human effort as well, because you only need to set up the learning once and train it once, and you can produce parameter fields for the entire USA! We were never able to do this easily in the past! Again, obviously, this contrast depends on how many sites you are training together (Figure 5 from Tsai et al., 2021).

[Figure]

**Fig. 5 Comparison of parameters generated by dPL and SCE-UA.** The continuous, spatially representative patterns of **a** dPL-inferred parameters are noteworthy, especially in comparison to the discontinuous, random appearance of **b** SCE-UA-inferred parameters from site-by-site calibration. Both were trained with a $1/8^2$ sampling density.

In fact, the more sites that participate in training, the more robust (higher quality) the model becomes. There is a beneficial scaling relationship that is very alien to us geoscientists. We explored this in Tsai et al., 2021 as well.

[Figure]

**Fig. 7 Scaling curves of dPL. a** Training data sampling density increases from s16 to s4. Dashed and solid red curves share the same red y axis (dashed line is for SCE-UA, to enable comparison). **b** Scaling curve for the spatial extrapolation (PUB) test with the Beck20 global headwater catchment dataset.

Wen-Ping Tsai*, Dapeng Feng*, Ming Pan, Hylke Beck, Yuan Yang, Kathryn Lawson*, Jiangtao Liu*, and Chaopeng Shen. From calibration to parameter learning: Harnessing the scaling

effects of big data in geoscientific modeling. *Nature Communications*, (2021). doi: 10.1038/s41467-021-26107-z.

L200-205: Employing FHV and FLV is a thoughtful model evaluation strategy. In contrast to relying solely on integrated metrics like KGE, FHV and FLV offer a more thorough evaluation of model performance. Has the author considered whether integrated metrics such as KGE are suitable for capturing the distribution characteristics of state variables? Additionally, is the loss function used during the neural network parameter determination suitable for the data's distribution characteristics? This is crucial, as KGE includes a term related to Pearson correlation coefficient, which may not be applicable to distributions beyond normal. Such considerations are essential to avoid potential misguidance in the model calibration process.

We will add some comments regarding this point in the revision. In our experience, KGE tends to emphasize the peaks even more than NSE. Please note that KGE was used as an evaluation metric in the paper but the main calibration objective was not KGE --- it was a mixture of mean squared error and root-mean-squred error, as described in the paper. We are aware of the distribution issue raised by the reviewer. As described in an earlier paper (Feng et al., 2020), we applied a logarithmic transformation which made the residual more normal. However, later on this was found to be not as crucial as we initially thought, because NNs are good at minimizing any errors that is minimization and the tradeoff is not that strong.

it should be a separate paper that discuss the selection of metrics, for example, see this one https://hess.copernicus.org/articles/23/4323/2019/.

L208-212 & Figure 3: Are the evaluation results corresponding to the training set, the testing set, or the overall results from both? The assessment outcomes should be provided separately for the training set and the testing set to enable a clearer evaluation of the model's performance and identification of any issues within the model.

Metrics are reported for the validation data. We will clarify this.

L222-223: The structural issues might explain the errors with peak flow in differentiable models. However, it raises a question as to why LSTM also exhibits errors in peak flow. As mentioned earlier, the utilization of inappropriate evaluation metrics could contribute to the low FHV across all models. It requires a more in-depth consideration of how metrics impact the calibration results.

We will add some comments in the revised manuscript. Our hypotheses are (i) underestimation of precipitation inputs especially for large storms; (ii) to capture extremes, we need smaller basins and probably run models on hourly time steps to capture the spatial heterogeneity in rainfall and nonlinear effects; (iii) extremes are not well observed so the training of NNs may be problematic for extremes; (iv) as you mentioned, some consideration with calibration metrics as well, but some previous effort including from Kratzert et al., have studied this pretty extensively

and they were not able to find a way to improve extreme metrics. We think much more effort is needed to improve extreme representation and we welcome community effort in study this topic.

---

## Author Response (AR1)

Dear GMD Editor(s),

Thank you for handling the manuscript. We have completed a round of revisions. Most of the comments are asking for clarification, except for the first point of RC1 who asks about the novelty of the work. We must say that although the model structure has not changed from the last paper, there are substantial uncertain questions regarding effectiveness, generalizability, and strengths and limitations when differentiable models and LSTM are applied to the global scale. We gained lots of insights from this paper, including what state-of-the-art prediction accuracy these models can achieve at the global scale by learning from large data; how different models behave in global catchments across varied climate groups and how their performance patterns can be related to geographic characteristics and hydrologic processes; what physical representations need to be improved towards developing stronger global models; and which modeling approach may be more appropriate to serve as next-generation global models to characterize the hydrologic processes in ungauged regions across the globe.

None of these findings would have been clear unless we attempted to apply these models at the global scale. Hence we think this work has suitable value for the community. This paper was classified as a Model Evaluation paper which we think appropriately captures the type of the study. It also serves as an important benchmark and model diagnosis for future global hydrologic modeling.

**RC1:**

This study presents a simulation using a differentiable model at 3753 basins globally. It represents an advance in the field and is thus worthy of being published somewhere. However, the used model and the design of the experiments are almost the same as those published in the authors' previous papers. Readers of GMD would expect more progress in those aspects. Thus, I suggest a major revision.

We thank you for your comments. We would like to note, although the main model structure has not been changed, the dataset, training and related experiments changed drastically. We expanded the dataset from only the USA to the whole globe with much larger numbers of basins with diverse climate and geographic attributes, which represents a substantial amount of work. Enormous efforts went into cleaning the data, improving model efficiency, training and testing, and interpreting the results. It was highly uncertain if the model qualities shown at CONUS scale would carry to such a large dataset at global scale. Many of the hydrologic insights are new. The novel experiment cross-continent predictions examined if differentiable models are strong candidates for solving the practical issue of prediction in ungauged regions at global scale. Plus, we do want to ensure some consistency between the evaluation across datasets so we can understand how the same model structure behaves on different datasets. We revised the manuscript to better highlight our novel research questions, as shown below. Please also see our responses to other comments to address this point.

This paragraph is modified to explain the progress of this study to our previous studies,

*"We desire efficient regionalized models that maximally leverage available information and provide accurate predictions to diverse basins across different climate groups and geographic characteristics in the world. We also want the models to perform decently even in data-sparse regions, showing competitive extrapolation ability, given that many large regions such as in Africa and Asia lack publicly available streamflow data. DL and differentiable models seem plausible candidates for such simulations. Nevertheless, previous studies on DL and physics-informed differentiable models mainly focus on continental or smaller scales, with a relatively homogeneous forcing dataset --- it is unclear if their observed strengths, e.g., high performance and strong generalization ability, can carry over to global scales, where the climate is much more diverse and datasets differ widely in their biases and uncertainty characteristics. In particular, we want to thoroughly examine how well these models can leverage information learned in data-rich continents to characterize the hydrologic processes in ungauged regions across the world. Meanwhile, DL models also show favorable scaling relationships (or data synergy) where more data leads to more robust models (Fang et al., 2022). Thus, training on a larger dataset may provide additional benefits."* lines 97-108

We also added these statements and discussions to clarify this,

*"We first investigate what prediction accuracy can be achieved by different models at global scale by learning from a large and diverse dataset. We then relate the global spatial patterns of model performance to geographic characteristics and hydrologic processes to identify model structural deficiencies and gain hydrologic insights. Finally, we provide evidence indicating which type of model may be more appropriate for next-generation global modeling by rigorously examining their generalizability to ungauged regions across the world."* lines 117-121

*"This study builds a benchmark and a basis for model selection and diagnosis for the next-generation global hydrologic modeling, which previously did not learn from such large observations. With rigorous tests at global scale, this study proves that differentiable models are strong candidates as global water models. With powerful spatial generalization ability, they can be applied to characterizing the hydrologic processes in ungauged regions by leveraging learned information in data-rich continents. Differentiable models in this study have already learned the generic and robust relationships between geographic attributes and physical model parameters from thousands of global catchments. Therefore, these models can be easily applied towards providing seamless global hydrologic modeling with parameters directly generated from worldwide geographic attributes. Future work can use such models to produce global hydrologic fluxes while enhancing some process representations in extremely arid, glaciated, or heavily human-influenced basins."* lines 427-435

GMD specifically has a model evaluation paper category, and there are many papers on GMD which evaluate the same model under different scenarios. In fact, we can probably say that it is a minority case to be changing the model structure significantly from paper to paper. As such, we

think the paper fits GMD very well, and will be a valuable contribution to the model evaluation literature.

Major comments

- The authors should introduce more details of the experimental design, such as the metrics used in the training, how many experiments (temporal generalization, PUB, and PUR; correct me if I am wrong), and the purpose of the experiments. Some details may have been presented somewhere else. I find this manuscript difficult to follow without reading the authors' previous publications.

Thanks for the comment. We have revised the manuscript to give some more background on earlier work, and added details for metrics used, experimental design, etc.

*"As in Feng et al., (2022), we employed the loss function based on root-mean-square error (RMSE) with two weighted parts. The first part calculates RMSE directly on the simulated and observed discharge, while the second part calculates RMSE on the transformed discharge records to improve low flow representations."* lines 179-182

Section 2.4 explains all the experiments and and we modified it as,

*"We ran one temporal and two spatial generalization experiments to evaluate the performance of different regionalized models. For the temporal generalization experiment, the models were trained for the period of 2000 to 2016 on all global basins, and tested for the period of 1980 to 1997. Without spatially holding out any basin during training, this experiment aimed at evaluating the model's generalizability in the time dimension by testing prediction ability on the same basins but in a different time period from the training data. The other two spatial generalization experiments served as the true litmus tests for evaluating the effectiveness of regionalization schemes, i.e., how well the model can be applied to basins that have never been seen during training. The first spatial generalization experiment was a traditional "prediction in ungauged basins" (PUB) problem, where we randomly divided the whole global basin set into 10 folds (groups) and performed cross-validation across these folds to obtain spatial out-of-sample predictions for all basins (training on 9 of the folds with the 10th fold held out and testing on the 10th, then rotating such that each fold is used for testing once). The second spatial generalization experiment, which we refer to as cross-continent "prediction in ungauged regions" (PUR), was more challenging. In this experiment, we assumed that all the basins in certain continents are ungauged and excluded from the training dataset, trained a regionalized model in other data-rich continents, and then tested the trained model to make predictions in the ungauged continents. With random hold-out, an ungauged test basin in the first spatial generalization experiment always has training gauges surrounding it. Therefore, the first PUB experiment can be interpreted as spatial interpolation. The second spatial experiment (cross-continent PUR) holds out all the basins in one continent as testing targets, and thus is the*

*much harder test of spatial extrapolation."* lines 202-218

We also clarified that *"All the reported performance metrics in this study are from model evaluation on the testing dataset, which is not seen by the model during the training process."* lines 225-227

- L124: what are the criteria for the selection? Can you describe the erroneous cases? Do the erroneous cases include the data processing error described in L350?

There are some data records that just seem totally unreasonable, for example, total discharge records are way larger than precipitation, yet unable to be linked to some simple unit conversion errors. Some basins also have discharge values that show differences of several magnitudes for two time periods. Line 124 and Line 350 in the original manuscript refer to the same group of gages with records showing these phenomena. We excluded those basins with significant issues but kept those gages showing moderate issues. We now clarified this statement as *"We excluded some basins with potential erroneous discharge records such as showing unreasonable magnitude way larger than precipitation or dramatic differences between two time intervals, by manually performing visual screening"*. Lines 132-134

- L306-L317: Can you discuss more about comparing the traditional regionalization method and PUR? The PUR regionalization method can utilize a large number of observations to calibrate/train the differentiable model, whereas the traditional method can only use very limited samples. In other words, PUR may have a much higher chance of finding the optimal parameters than the traditional method.

Here we compare traditional parameter regionalization methods with differentiable modeling. PUR and PUB, as spatial generalization tests, are the experiments to test the performance of different regionalization methods. In this study, differentiable models largely outperform the traditional method built in Beck et al., 2020 that represents the previous best spatial generalization results at global scale. It has also been established in past work that differentiable models have better performance than another traditional regionalization scheme, MPR (see the large gap between MPR+mHM and all the different models in the below figure from Feng et al., 2022). These traditional methods actually have a similar concept to differentiable modeling, which aims at building the relations between geographic attributes and model parameters, but the traditional methods only use linear functions because the model is not differentiable and they cannot optimize a large number of weights (as in neural networks) like the differentiable models can. Differentiable models instead don't assume any linear transfer function forms for the relations, and automatically learn the complicated relations by the embedded neural networks from big data. Hence what the reviewer said is true, but we would modify it a bit as "differentiable models have a much higher chance of finding the most high-performing, robust, and transferable parameters compared to the traditional method." We added these statements to incorporate the discussion,

*"Differentiable modeling does not rely on strong assumptions of the functional form for the parameter transfer function. It leverages the powerful ability of neural networks to represent complicated functions, and automatically learns robust and generalizable relationships between geographic attributes and physical model parameters from large data. Therefore, we can expect significant performance advantages from differentiable modeling compared to traditional methods relying on linear transfer functions."* lines 336-340

(a) (b)

| | |
|---|---|
| **MPR+mHM**
 B=492 NSE$_{50}$: 0.528 | **dPL+evolved HBV ($\delta_n$)**
 B=671 NSE$_{50}$: 0.714 |
| **dPL+HBV ($\delta_1$)**
 B=492 NSE$_{50}$: 0.618 | **dPL+evolved HBV with DP ($\delta_n(\beta^t, \gamma^t)$)**
 B=671 NSE$_{50}$: 0.732 |
| **dPL+HBV ($\delta_1$)**
 B=671 NSE$_{50}$: 0.64 | **LSTM**
 B=671 NSE$_{50}$: 0.748 |

Minor comments

- Title: the phrase, global hydrologic simulations, is misleading. The simulations are conducted at 3753 basins across the globe. It represents a concept different from the "global hydrologic simulations."

We revised the title to *"Deep dive into hydrologic simulations at global scale"* to avoid confusion. However, it's worth emphasizing that differentiable models shown in this study have already learned the relationships between geographic attributes and model parameters at global scale by learning from thousands of catchments. This means the models can be easily applied for seamless global modeling with parameters directly generated everywhere from geographic attributes using the trained models, which will be the subsequent work following this study. We can imagine that for global seamless simulations, the largest barrier would be generating reliable parameter fields, which is largely solved in this study. We also added these statements to further clarify it,

*"Differentiable models in this study have already learned the generalizable and robust relationships between geographic attributes and physical model parameters from thousands of global catchments. Therefore, these models can be easily applied towards providing seamless global hydrologic modeling with parameters directly generated from worldwide geographic attributes. Future work can use such models to produce global hydrologic fluxes while enhancing some process representations in extremely arid, glaciated, or heavily human-influenced basins."* Lines 431-435

- L127: is the classification from Beck et al., 2020b?

Yes. We modified the statement as *"These basins had been classified into five Köppen-Geiger climate classes in Beck et al., (2020b), including tropical (489 basins), arid (109 basins), temperate (1423 basins), cold (1593 basins), and polar (139 basins), as shown in Figure 1"*. Lines 136-138

- L26 & L212: How is the subset of 1675 basins selected? What is the objective of the selection?

The selection criteria was long-term streamflow data availability. Here, "long-term" means that these basins have at least 15 years' worth of records (not necessarily continuous) in the training time period and 5 years' worth of records in the testing period for model evaluation. We have modified the following statement in the main text,
*"For a subset of 1675 basins with long-term records (at least 15 years' worth of streamflow data available in the training period and 5 years' worth of data available in the testing period, though not necessarily continuous), LSTM even reached a KGE of 0.78."* Lines 234-236

- L223: can you describe more about the structural issues? Why does the explicit solution scheme introduce numerical errors here?

Here we mentioned numerical errors introduced by the explicit scheme. There are also issues related to how groundwater storage cannot feed back to the upper layers. We modified the original statement as *"the errors with peak flow could also be partly due to some numerical and structural issues with the differentiable models, e.g., numerical errors introduced by the explicit and sequential solution scheme of HBV with excessive use of threshold functions that lead to different results when the sequence changes, and structure limitations, e.g., deeper groundwater storage cannot feed back to the upper layers."* Lines 248-251

- L291-L293: please rewrite the sentence. It is difficult to read.

We revised the statement to *"The sensitivity of model performance to missing processes in the differentiable models is both good and bad news. It is good news because this means we can identify suitable or insufficient process representations by learning from data..."* lines 318-320

- L398, "the underrepresentation of the processes...": this conclusion is too general. The difficulty of representing arid/polar/anthropogenic processes is known before reading this paper. The conclusion should be specific.

There are many structural deficiencies with HBV, but it seems the neural network based parameterization approach in differentiable modeling can make up for some of the deficiencies. Here we are saying that in these two cases, even the NN-based parameterization cannot make up for these missing processes. We largely clarified these statements as the following:

*"For the polar group, the differentiable model performed significantly worse than the LSTM. Without any physical constraints, LSTM shows strong power in simulating storage (snow and glacier) dominated processes, while differentiable models are limited by the structure of their physical backbone model, which in this case does not simulate multiyear ice buildup and melt. Another limitation could be soil sealing processes in extremely arid regions. These regional performance comparisons thus reveal some deficiencies of the physical backbone in δHBV that cannot be mitigated even by advanced neural network-based parameterization."* Lines 448-454

**CEC1:**

Dear authors,

I have checked the "Code and Data Availability" section in your manuscript. I would like to point out that the sentence saying that an updated version of the code and data will be available upon acceptance is misleading. Obviously, if your manuscript is accepted for publication, you have to publish it with the most updated code and data, and we do not accept "upon acceptance" statements. Right now, your statement seems to imply that you have not published your code and data in advance, however, they are included in the Zenodo repository. I would like to ask you to modify the sentence to avoid confusion.

Also, please clarify which is the PyTorch version that you have used for your work.

Regards,

Juan A. Añel

Geosci. Model Dev. Executive Editor

Thanks for the notice. The models themselves were indeed published previously. They are applied to a new (larger) global dataset to yield new insights, since they were not previously applied on the global scale. We have modified it in the revised manuscript along with clarifying the PyTorch version as shown below.

*"...was implemented in a DL platform (PyTorch 1.0.1 was used for the original development and the model has also shown good compatibility with more recent PyTorch versions, (Paszke et al., 2017))..."* Lines 176-178

*"The source codes for the differentiable hydrologic models can be accessed at https://doi.org/10.5281/zenodo.7091334, and this study evaluates these models at global scale. "* Lines 473-475

**RC2:**

Overall, the manuscript is well-written with a clear research objective, innovative hybrid models, and solid results. The study compared the performance of the commonly used LSTM model with two differentiable hydrological models (static and dynamic parameters) with a temporal generalization experiment and conducted a comparative analysis for traditional "prediction in ungauged basins" problem. The model evaluation results provide valuable insights into improving the mechanism of the hydrologic models, confirming the strong localization and extrapolation capabilities of differentiable models. Below are some key comments and concerns:

Thank you very much for the evaluation. Please see our responses to each of the comments below.

L161-166: From my understanding, the original parameter calibration process of the HBV model has been replaced by the parameter calibration process of the *gA* neural network. If this is the case, it is still necessary to run the HBV model. How does this approach compare to the traditional hydrological model calibration methods in terms of modeling speed? Has it resulted in time and labor savings in the modeling process?

Good question. The HBV model has been reimplemented on the PyTorch platform, gaining the ability to track gradients and run in high concurrency on GPUs. To accommodate high concurrency, we had to reimplement the code in a parallel way. We further require that a generic parameterization network is trained over all the thousands of sites rather than being calibrated site-by-site, learning a robust mapping relationship. This differentiable modeling framework resulted in orders of magnitude of computational savings, and we'd argue labor saving as well, but it depends on the economy of scale. Below we provide some more background. We realize that many people have not read our previous paper (Tsai et al., 2021), and we include some pieces of the discussions below in the revised manuscript.

There is an obvious time saving in parallelism. During training, we run models in parallel over a "minibatch", which means integrating data from 256 basins at a time, before we calculate the loss and update the weights. This efficient parallelism is due to easy GPU concurrency via PyTorch.

There is a less obvious economy of scale, which is that the training and the learned knowledge of neural networks are shared by all sites. This means the more sites participating in the training, the more efficient it becomes (see the figures below, from Tsai et al., 2021). Hence, if you run differentiable model training for one of a few sites, it won't be as efficient as traditional optimization. If you run it for many more sites, it will become orders of magnitudes more efficient. The number we cite in Tsai et al., 2021 is that, for the same calibration job over the USA, a traditional evolutionary algorithm would need a 100-processor cluster to run 2-3 days, whereas our differentiable parameter learning method only needs 1 single GPU for one hour for a surrogate model. Similarly, only several hours are required for the differentiable HBV with 16 components (the original HBV does not have this subbasin-scale heterogeneity).

[Figure]

More importantly, this differentiable model can achieve several things for which the computational cost would be impossible if using a traditional calibration approach. We have flexibly evolved the structure of the original HBV in the differentiable modeling approach by using multi-component computation and dynamic parameters, as stated in line 177 of the original manuscript. The first version increases HBV parameters from 12 to 12*16=192 with 16 components and the second version has a different parameter value at each time step, varying with the meteorological forcings. It's unrealistic to use traditional calibration to achieve the optimization for these models. However, all these static and dynamic HBV parameters can be automatically and efficiently learned with differentiable modeling.

We argue it actually saves human effort as well, because differentiable modeling enables robust parameter regionalization, which means it optimizes and learns the relations between catchment geographic attributes and model parameters. We only need to set up and train the model once and then it can generate parameter fields across the world and be applied to prediction in ungauged regions as shown in this manuscript. We were never able to do this easily in the past.

We added these discussions to the main text,

*"Some central differences exist between the differentiable modeling approach and the traditional calibration approach: First, we always attempt to train differentiable models over a large*

*collection of sites, improving the robustness and efficiency of learning. Second, we reimplemented the model onto PyTorch in a parallel fashion, and were thus able to leverage the high concurrency and computing power offered by modern GPUs. Third, the commonalities between sites and the accumulation of knowledge in the neural network further improves the efficiency of training compared to traditional training done individually for each site, as the knowledge learned from one batch is inherited when the neural network is trained on the next batch. Fourth, differentiable models can flexibly evolve the structure of the physical backbone. The two types of differentiable models used in this study have used multiple parallel components per basin to represent subbasin-scale heterogeneity. For the models with dynamic parameterization, two parameter values vary at each daily time step as a function of the meteorological forcings. It seems unrealistic to use traditional parameter calibration to optimize these models with evolved structures. However, leveraging automatic differentiation and gradient descent, differentiable models can automatically learn to produce large amounts of parameters from geographic attributes through the embedded neural networks.* Lines 393-404

*"This study builds a benchmark and a basis for model selection and diagnosis for the next-generation global hydrologic modeling, which previously did not learn from such large observations. With rigorous tests at global scale, this study proves that differentiable models are strong candidates as global water models. With powerful spatial generalization ability, they can be applied to characterizing the hydrologic processes in ungauged regions by leveraging learned information in data-rich continents. Differentiable models in this study have already learned the generic and robust relationships between geographic attributes and physical model parameters from thousands of global catchments. Therefore, these models can be easily applied towards providing seamless global hydrologic modeling with parameters directly generated from worldwide geographic attributes. Future work can use such models to produce global hydrologic fluxes while enhancing some process representations in extremely arid, glaciated, or heavily human-influenced basins."* Lines 427-435

Tsai, W. P., Feng, D., Pan, M., Beck, H., Lawson, K., Yang, Y., ... & Shen, C. (2021). From calibration to parameter learning: Harnessing the scaling effects of big data in geoscientific modeling. Nature communications, 12(1), 5988. doi: 10.1038/s41467-021-26107-z.

L200-205: Employing FHV and FLV is a thoughtful model evaluation strategy. In contrast to relying solely on integrated metrics like KGE, FHV and FLV offer a more thorough evaluation of model performance. Has the author considered whether integrated metrics such as KGE are suitable for capturing the distribution characteristics of state variables? Additionally, is the loss

function used during the neural network parameter determination suitable for the data's distribution characteristics? This is crucial, as KGE includes a term related to Pearson correlation coefficient, which may not be applicable to distributions beyond normal. Such considerations are essential to avoid potential misguidance in the model calibration process.

We used multiple metrics such as KGE, NSE, FLV and FHV to comprehensively evaluate model performance, but KGE was not the main objective function used to train the models in this study. The loss function was defined as a mixture of two parts based on the root-mean-square error (RMSE). We are aware of the distribution issue raised by the reviewer and thus employed this loss function. As shown in our previous study and the below equation (Feng et al., 2022), we applied a logarithmic transformation on the discharge data which made the residual more normal, and then further calculated RMSE on the transformed data as one part of the loss function. This part aims at improving the low flow representation. The second part is RMSE directly calculated on the discharge data, and we did a weighted combination of these two parts. We previously tested using MSE, RMSE, NSE as loss functions and they produced quite similar evaluation results, which implies the selection of calibration objectives was not as crucial as we initially thought. However, this weighted two-parts RMSE with data transformation applied can largely mitigate the issue of peak flow being over-emphasized from using the above-mentioned objective functions.

$$\widehat{Q}^t = Log_{10}(\sqrt{Q^t} + 0.1)$$

We added these statements in the text to clarify the loss function used to train the models:

*"As in Feng et al., (2022), we employed the loss function based on root-mean-square error (RMSE) with two weighted parts. The first part calculates RMSE directly on the simulated and observed discharge, while the second part calculates RMSE on the transformed discharge records to improve low flow representations."* Lines 179-182

Feng, D., Liu, J., Lawson, K., & Shen, C. (2022). Differentiable, learnable, regionalized process‐based models with multiphysical outputs can approach state‐of‐the‐art hydrologic prediction accuracy. Water Resources Research, 58(10), e2022WR032404.

L208-212 & Figure 3: Are the evaluation results corresponding to the training set, the testing set, or the overall results from both? The assessment outcomes should be provided separately for the training set and the testing set to enable a clearer evaluation of the model's performance and identification of any issues within the model.

All the evaluation results reported in this manuscript are for the independent **testing dataset**. We completely agree that this is very important for fair model evaluation. We added and modified the following statements to clarify how we evaluate the model performance and report metrics.

*"All the reported performance metrics in this study are from model evaluation on the testing dataset, which is not seen by the model during the training process."* Lines 225-227

*"For the temporal generalization experiment, the models were trained for the period of 2000 to 2016 on all global basins, and tested for the period of 1980 to 1997. Without spatially holding out any basin during training, this experiment aims at evaluating the model's generalizability in the time dimension by testing it in a different time period. The other two spatial generalization experiments serve as the true litmus tests for evaluating the effectiveness of regionalization schemes, i.e., how well the model can be applied to basins that have never been seen during training."* Lines 203-208

L222-223: The structural issues might explain the errors with peak flow in differentiable models. However, it raises a question as to why LSTM also exhibits errors in peak flow. As mentioned earlier, the utilization of inappropriate evaluation metrics could contribute to the low FHV across all models. It requires a more in-depth consideration of how metrics impact the calibration results.

Our hypotheses for the causes of peak flow bias even in LSTM are (i) underestimation of precipitation inputs especially for large storms; (ii) to capture extremes, we need smaller basins and probably will need to run models on hourly time steps to capture the spatial heterogeneity in rainfall and nonlinear effects; (iii) extremes are not well observed so the training of neural networks (NNs) may be problematic for extremes; (iv) as you mentioned, some consideration with calibration metrics as well, but some previous efforts including from Kratzert et al., have studied this pretty extensively and they were not able to find a way to largely improve extreme metrics. We must mention that all these studies with different models and calibration metrics have shown the underestimation of peak flow (please see the FHV column in Table 3 in Kratzert et al., 2019), while integrating forcing data from several sources could moderately mitigate this underestimation (Kratzert et al., 2021), suggesting the forcings to be an important source for this issue. Therefore, the effects of the last hypothesis on the calibration metric could be limited, especially given the use of MSE, RMSE, NSE; these mean-square based metrics already emphasize large peak flow values. We think much more effort is needed to improve extreme representations and we look forward to community effort and collaboration in studying this topic.

We modified these statements in the text to discuss this point:

*"However, for the peak flow predictions, the LSTM and differentiable models were quite similar, and they all underestimated the observed peaks (FHV in Figure 3). The underestimation for peak flows is consistent with what was found in previous studies. For example, all the physical and deep learning models have significant negative peak flow bias when benchmarked in the CONUS dataset (Feng et al., 2020; Kratzert et al., 2019b). We hypothesize that the systematic underestimation of peaks may be partially related to bias in precipitation forcings. MSWEP is based on the ERA5 reanalysis, which is known to underestimate precipitation peaks (Beck et al.,*

*2019). Furthermore, the use of basin-averaged, daily-averaged precipitation may further suppress the peaks (Chen et al., 2017). In addition, the errors with peak flow could also be partly due to some numerical and structural issues with the differentiable models, e.g., numerical errors introduced by the explicit and sequential solution scheme of HBV with excessive use of threshold functions that lead to different results when the sequence changes, and structure limitations, e.g., deeper groundwater storage cannot feed back to the upper layers. Given the commonality of this issue, we call for community efforts and collaboration to address this issue."* Lines 242-252

Kratzert, F., Klotz, D., Shalev, G., Klambauer, G., Hochreiter, S., & Nearing, G. (2019). Towards learning universal, regional, and local hydrological behaviors via machine learning applied to large-sample datasets. Hydrology and Earth System Sciences, 23(12), 5089-5110.

Kratzert, F., Klotz, D., Hochreiter, S., & Nearing, G. S. (2021). A note on leveraging synergy in multiple meteorological data sets with deep learning for rainfall–runoff modeling. Hydrology and Earth System Sciences, 25(5), 2685-2703.

---

## Referee Report (RR1)

**General comments**

Feng et al tested the performance of a hybrid model in simulating daily streamflow at several thousand global watersheds. They benchmarked the performance of the hybrid model with a deep learning model and found the hybrid model has a satisfied performance. The topic is a good fit to the scope of Geoscientific Model Development, though I agree with previous review that this study uses the same model and design of experiments from the authors' previous study. The new insight of this work is to test the hybrid model framework in more watersheds locate from different continents. The idea of hybrid model is great and represents a significant contribution to hydrological modeling. It would be helpful for the authors to discuss the challenges of coupling the ML/DL model with a hydrological model. Specifically, since HBV model is relatively simple, is it possible to develop such a hybrid model with a more complicated hydrological model? Please also see my specific comments in the following.

**Specific comments**

As the author argued in Line 47 – Line 55 that DL and LTSM have been demonstrated with good performance of simulating hydrological variables from local to continental scales, I wonder what is the novelty of this work? Although this study extends previous application at CONUS to global scales, the selected dataset doesn't have a good coverage for the whole globe. Please find my detailed comments regarding the selection of dataset in the following.

Line 124: Why the authors select this dataset given there exist other global streamflow datasets that contain much more gauges and have a better spatial coverage? Since the selected dataset archives headwater catchments, the observed streamflow is approximately same as the runoff generation, with the less river routing impacts. Is this because there is not river routing component in HBV? If runoff is the target variable, there are multiple well validated global runoff datasets to be used. Then it is not reasonable for the author to justify that the application is at global scales. In addition, Figure 1 shows the selected catchments are mainly from certain regions. Except North America, the gauges over other continents do not cover the continent uniformly, thus they are not representative for the continent.

Line 95 – Line 97: There are several studies that calibrated the models to match monthly variations.

Section 2.3: In traditional hydrological model calibration, one needs to run the physical model many times with perturbed parameters. How many forward simulations are needed in the hybrid model to identify the best parameter? Please clarify.

Line 202 – Line 218: I agree with the authors that PUR experiment is more challenging spatial extrapolation than PUB experiment. However, in practice, PUB experiment is more useful than PUR. Streamflow is probably the most well observed hydrological variables. At global scales, there exists abundant streamflow gauges with a good spatial coverage for each continent, though some continents have relative less than others. Therefore, one doesn't need to assume a whole continent is ungauged. It is the selected dataset in this study that gives sparse spatial coverage.

Line 236: Do you mean a median KGE of 0.78?

Section 3.2: I expect dPL + evolved HBV with DP is always better than dPL + evolved HBV, because the former model is more flexible to capture the observation. If no better, it should not be worse than the latter one. But Figure 5 shows, dPL + evolved HBV with DP is worse than dPL + evolved HBV in arid region. It will be helpful for the authors to clarify such clarification.

Line 341: I don't think the median KGE = 0.58 is significantly better than median KGE = 0.52. They are pretty close performance to me. I suggest the authors to plot the cumulative density functions (CDFs) of the KGE for different model, and test if the CDFs are statistically different.

Line 350 – Line 351: Do you mean "cannot be obtained by straightforwardly training *physical* or *DL* models on data alone"? Figure 6 suggests the hybrid model is better than the traditional model calibration (Beck20). But it doesn't support that performance of hybrid model is statistically better than a purely data-driven model.

Line 371: Why not using KGE and NSE for the ET evaluation to be consistent with discharge evaluation?

---

## Author Response (AR2)

Dear Editor,

We appreciate your patience in receiving these changes, as the first author now has changed institutions and has additional responsibilities. We appreciate the constructive feedback from you and both reviewers, and hope the following changes address your comments and concerns. Line numbers refer to the manuscript version with tracked changes.

Reviewer #1
The authors have indicated that this paper maintains the same model structure as a previous publication by the same research group. However, considerable effort has been invested in preprocessing the data and integrating it into the model pipeline, leading to the discovery of novel insights. Despite this, while the paper meets the criteria for publication and possesses scientific significance to a certain extent, it leans more towards generating insights from a new application rather than serving as a model evaluation paper. Consequently, I believe that this paper may find a better fit in a journal such as Water Resources Research, which aligns more closely with the findings presented. Therefore, I recommend that the editor reconsider whether to accept the paper in its current form or suggest that the authors submit it to a more suitable journal.

Thanks for your comments! GMD has a paper type called model evaluation which we think this paper fits quite well in. Many of the past papers published in this category evaluate an existing model structure under different contexts. As the reviewer acknowledged, a large amount of work was needed to run the model on the global scale.

Deep learning-based methods and the increasing amounts of observations provide great opportunities for global hydrologic modeling. In this paper, we evaluated both purely data-driven and physics-informed deep learning models globally on thousands of catchments. These models all support parameter regionalization, thus can be flexibly applied to predictions in ungauged regions over the world. Although spatial generalization is vital to global-scale modeling, few studies have systematically examined this issue with state-of-the-art deep learning and physics-informed deep learning methods at such a large scale as shown in this study. We also distill global hydrologic insights from the model evaluation to support future development and structure modification. This study is a necessary and important step to support the model selection for next-generation global models and perform highly accurate seamless modeling covering the global data-sparse regions. Therefore, we believe this study fits quite well to the scope of GMD for the model evaluation category.

Reviewer #2
General comments
Feng et al tested the performance of a hybrid model in simulating daily streamflow at several thousand global watersheds. They benchmarked the performance of the hybrid model with a deep learning model and found the hybrid model has a satisfied performance. The topic is a good fit to the scope of Geoscientific Model Development, though I agree with previous review that this study uses the same model and design of experiments from

the authors' previous study. The new insight of this work is to test the hybrid model framework in more watersheds locate from different continents. The idea of hybrid model is great and represents a significant contribution to hydrological modeling. It would be helpful for the authors to discuss the challenges of coupling the ML/DL model with a hydrological model. Specifically, since HBV model is relatively simple, is it possible to develop such a hybrid model with a more complicated hydrological model? Please also see my specific comments in the following.

Thanks for the comments!

We added the following paragraph to discuss how to implement a more complicated differentiable model and its challenges.

*"To create a new differentiable model or turn an existing model into a differentiable one, we need to implement the model on a differentiable platform like PyTorch, Tensorflow, or JAX, while better enabling model parallelism in order to maximally leverage the computing power of modern graphical processing units (i.e., GPUs). If a model contains mostly explicit calculations, automatic differentiation (AD) offered by the above platforms can effortlessly provide gradient calculations, requiring only a syntax-level translation which can nowadays be done easily. Sometimes, a limited amount of adjustments are needed to turn non-differentiable operations into equivalent differentiable ones. However, when a model contains iterative solutions to nonlinear systems, large matrix solvers or constrained optimizations, we can employ the adjoint method (Song et al., 2023). The adjoint method explicitly defines the gradient-calculation method and alters the order of calculations so iteration is avoided during gradient calculations, which can dramatically reduce memory demand and improve efficiency. Another important consideration is the effective use of parallelism and the modern computing infrastructure for AI (i.e., GPUs). In our context, the regionalized parameterization (in this case, training one neural network on a large amount of basins), which is crucial to ensuring the generalizability of the model, requires going through large data in high-throughput parallelism. Embracing parallelism may necessitate some coding adjustments. At this point, several versions of differentiable hydrologic models have been proposed with varying complexities and different handling of parameterization, post-processing (which we didn't use in this study, as it can interfere with interpretability of the internal variables, mass balances, and the sensitivity to inputs encoded by the process-based components) and dynamical parameters. Across geoscientific domains, differentiable ecosystem (Aboelyazeed et al., 2023; Zhao et al., 2019), streamflow and river routing (Bindas et al., 2024), water quality (Rahmani et al., 2023), and ice sheet (Bolibar et al., 2023) models have already been demonstrated." {Lines 408-426}*

This study mainly focuses on rainfall-runoff modeling, the key part of a more complicated hydrologic modeling system. We aim at identifying these two points from this study:

1) Which aspects of the used physical backbone should we improve next? Although the current framework provides strong performance advantages for large-scale modeling,

we identified several challenges from the evaluation process, such as the underrepresentation of processes like glaciers, ground ice, human interventions, etc. Future work should focus on improving these process representations in this modeling framework.

2) Does spatial generalization work for such a diverse global dataset with differentiable parameter regionalization? This is demonstrated by the spatial generalization tests which is very important to the global scale modeling given that many large lands across the world lack high-quality observations.

Beyond these two major aims, we can expect to integrate more physical processes in the future to develop the more complicated modeling system, as mentioned in the text: *"future work can gradually incorporate critical processes and include more observations to constrain the learning process, making sure each addition is valuable and accretive. The research community collectively has already substantial experience in evolving earth system models to include many processes. We expect some processes to be invited back in the differentiable modeling framework. Nevertheless, with differentiable modeling, we now have a new tool that was not previously available: highly flexible deep neural networks that can be placed anywhere in the model, which provide a systematic way of managing model complexity. With their help, such model evolution may take much less time than previously required. However, we still expect the development cycle to take longer than for purely data-driven models like LSTM, requiring us to view differentiable models as evolving rather than static entities, which need a bit of patience while maturing."* {Lines 453-461}

Several efforts of differentiable models have been developed or are under development as shown in the above added paragraph. We can potentially integrate these developments into the current framework to acquire more advanced modeling systems with better physical representations. Other groups in the community are also implementing complex land surface models on differentiable platforms. In the following two links (and screenshots), you can find other groups who are following up to do their differentiable models in hydrology or other fields. https://meetingorganizer.copernicus.org/EGU24/session/48163

[Figure]

Felipe Saavedra, Noemi Vergopolan, Andreas Musolff, Ralf Merz, Carolin Winter, and Larisa Tarasova ✉

HS3.5 | Explainable and hybrid machine learning in hydrology
▶ Orals Mon, 08:30  ▶ Posters on site Mon, 16:15

10:05–10:15 | EGU24-12574 | HS3.5 ★ | ECS | Highlight | On-site presentation
Analyzing the performance and interpretability of hybrid hydrological models ▶
Eduardo Acuna, Ralf Loritz, Manuel Alvarez, Frederik Kratzert, Daniel Klotz, Martin Gauch, Nicole Bauerle, and Uwe Ehret ✉

Coffee break

Chairpersons: Marvin Höge, Basil Kraft, Shijie Jiang

10:45–10:55 | EGU24-17842 | HS3.5 ★ | ECS | Highlight | On-site presentation
Deep learning based differentiable/hybrid modelling of the global hydrological cycle ▶
Zavud Baghirov, Basil Kraft, Martin Jung, Marco Körner, and Markus Reichstein ✉

10:55–11:05 | EGU24-6656 | HS3.5 ★ | ECS | On-site presentation
Exploring Catchment Regionalization through the Eyes of HydroLSTM ▶
Luis De La Fuente, Hoshin Gupta, and Laura Condon ✉

11:05–11:15 | EGU24-4768 | HS3.5 ★ | ECS | On-site presentation
HydroPML: Towards Unified Scientific Paradigms for Machine Learning and Process-based Hydrology ▶
Qingsong Xu, Yilei Shi, Jonathan Bamber, Ye Tuo, Ralf Ludwig, and Xiao Xiang Zhu ✉

11:15–11:25 | EGU24-12068 | HS3.5 ★ | ECS | On-site presentation
Hybrid Neural Hydrology: Integrating Physical and Machine Learning Models for Enhanced Predictions in Ungauged Basins ▶
Rajeev Shrestha, Bjarte Beil-Myhre, and Bernt Viggo Matheussen ✉

11:25–11:35 | EGU24-2850 | HS3.5 ★ | ECS | On-site presentation
Development of a Distributed Physics-informed Deep Learning Hydrological Model for Data-scarce Regions ▶
Liangjin Zhong, Huimin Lei, and Jingjing Yang ✉

11:35–11:55 | EGU24-262 | HS3.5 ★ | solicited | Virtual presentation
Differentiable modeling for global water resources under global change ▶
Chaopeng Shen, Yalan Song, Farshid Rahmani, Tadd Bindas, Doaa Aboelyazeed, Kamlesh Sawadekar, Martyn Clark, and Wouter Knoben ✉

11:55–12:05 | EGU24-16235 | HS3.5 ★ | ECS | On-site presentation
Flow estimation from observed water levels using differentiable modeling for low-lying rivers affected by vegetation and backwater ▶
Phillip Aarestrup, Jonas Wied Pedersen, Michael Brian Butts, Peter Bauer-Gottwein, and Roland Löwe ✉

12:05–12:15 | EGU24-20602 | HS3.5 ★ | ECS | Virtual presentation
Enhanced Continental Runoff Prediction through Differentiable Muskingum-Cunge Routing (δMC-CONUS-hydroDL2) ▶
Tadd Bindas, Yalan Song, Jeremy Rapp, Kathryn Lawson, and Chaopeng Shen ✉

12:15–12:25 | EGU24-11778 | HS3.5 ★ | ECS | On-site presentation

**Specific comments**
As the author argued in Line 47 – Line 55 that DL and LTSM have been demonstrated with good performance of simulating hydrological variables from local to continental scales, I wonder what is the novelty of this work? Although this study extends previous application at CONUS to global scales, the selected dataset doesn't have a good coverage for the whole globe. Please find my detailed comments regarding the selection of dataset in the Following.

We added these sentences to show the novelty of differentiable modeling to the LSTM model. *"Compared to the LSTM model which only outputs discharge simulations, differentiable models offer a suite of interpretable variables including ET, soil water, recharge, baseflow, etc., thus providing a comprehensive description for the hydrologic cycle and far better interpretability."* {Lines 406-408}.

There are indeed some data issues that cause the gaps in the global map. However, the dataset already represents the best efforts and, in fact, even more comprehensive than some alternative contributions such as the Caravan dataset. The most original dataset from coauthor Hylke Beck's effort includes in total 21955 streamflow stations from several data sources. Then several criteria were applied to exclude some catchments including 1) without daily streamflow records; 2) "non-reference" catchments in GAGES-II with significant human interventions; 3) areas smaller than 50 km2 or larger than 5000 km2; 4) too short with less than 5 years data; 5) visual screening for erroneous streamflow data. Finally, a dataset with ~4000 catchments was formed with these criterions applied, as shown in Beck et al., 2020 that gives more details on the dataset formation. We further applied one round of manual screening on this dataset to

exclude some stations with apparently unrealistic records (like abrupt change and magnitudes of large differences in two periods) in our study.

As shown by our response in the last round, going from CONUS to global scale will require lots of work. We need real experiments in the global catchments to evaluate the applicability of different models at this large scale, especially for their spatial generalizability. All the global datasets have shown very sparse data availability in some continents like Africa and Asia, and how to improve the prediction in these regions is a key issue for global models and discussed by the extrapolation experiments in this study. In addition, we also gain lots of new insights from the large global dataset with diverse climate and geographic groups.

Line 124: Why the authors select this dataset given there exist other global streamflow datasets that contain much more gauges and have a better spatial coverage? Since the selected dataset archives headwater catchments, the observed streamflow is approximately same as the runoff generation, with the less river routing impacts. Is this because there is not river routing component in HBV? If runoff is the target variable, there are multiple well validated global runoff datasets to be used. Then it is not reasonable for the author to justify that the application is at global scales. In addition, Figure 1 shows the selected catchments are mainly from certain regions. Except North America, the gauges over other continents do not cover the continent uniformly, thus they are not representative for the continent.

Please see above our response to the last comment. We assume the reviewer is referring to a dataset like Caravan. Actually, while they are called different names, the core discharge data of most of these datasets were derived from the same data sources like GRDC (see the below Figure). You can already notice the gaps in Asia and Africa at a first glance. Also, many stations have stopped providing data and don't have long-term high quality data to support the modeling evaluation. Not all these stations and records are really available to use in the modeling work. As mentioned by the previous selection criteria, ~4000 catchments are the final selection amount after applying these necessary criteria. We will go to the similar catchments finally even if applying other compiled datasets like Caravan.

[Figure]

There is indeed internal routing in the model that could work up to about 5,000 km2 with effectiveness depending on the region. The reviewer is correct in that large river routing was not included --- this will be addressed in future studies. Since runoff is essentially not observable at large scales, current runoff datasets are mainly based on simulated products which will introduce additional bias to the evaluation. One of the main future works will be using the outcomes from this evaluation study to provide a global seamless runoff product. It's also worth noticing that, although the training is in basin-lumped format, this differentiable modeling has enabled learning the transfer functions to parameterize the model so that it can be easily applied to simulating all the terrestrial lands across the world. Differentiable routing is ongoing work (Bindas et al., 2024) and will be expanded to global scale in ensuing pursuits.

Line 95 – Line 97: There are several studies that calibrated the models to match monthly Variations.

We added this statement to include the literature that uses monthly discharge variation in calibration. *"Only very limited studies attempt to calibrate global models on monthly discharge variations (Werth and Güntner, 2010)."* {Lines 98-99}

Section 2.3: In traditional hydrological model calibration, one needs to run the physical model many times with perturbed parameters. How many forward simulations are needed in the hybrid model to identify the best parameter? Please clarify.

Deep learning-based models are trained in mini-batches for forwarding and optimization. Each mini-batch consists of a small subset group of training time sequences from the whole training data . Within one epoch we iterate and forward the model in each minibatch, and there are 1293 iterations in total. One epoch is defined as all the data points are forwarded once in a probability

concept after iterating over many mini-batches. We train our differentiable models for 50 epoches in total.

We added these statements in the method section to further clarify this point:

*"Differentiable models are also trained in mini-batches that are formed in the same way as training the LSTM streamflow model. Within one epoch, differentiable models are forwarded and optimized over the randomly formed mini-batches until the iterations have used all the training data points. We train the differentiable models for 50 epochs in total."* {Lines 185-187}

Line 202 – Line 218: I agree with the authors that PUR experiment is more challenging spatial extrapolation than PUB experiment. However, in practice, PUB experiment is more useful than PUR. Streamflow is probably the most well observed hydrological variables. At global scales, there exists abundant streamflow gauges with a good spatial coverage for each continent, though some continents have relative less than others. Therefore, one doesn't need to assume a whole continent is ungauged. It is the selected dataset in this study that gives sparse spatial coverage.

As mentioned in our response to the previous comments, it's not specifically related to the selection of our dataset. The reviewer can refer to the GRDC map shown above ---- lots of significant data gaps considering many gauges stopped reporting 20 years ago! If there were dense enough river gauge monitoring, there wouldn't have been satellite missions like SWOT --- which still won't fully fill the gaps due to its large-river nature. The alternative datasets share the same original sources and have the same issue that available streamflow records are quite sparse to use in some continents like Africa and Asia.

PUR is a general representation for the spatial extrapolation prediction, while PUB doing random hold-out is actually spatial interpolation, always having neighboring gauged basins as representative donors. Therefore, PUR is a more practical issue in the real world, because in most real cases, the whole large continuous regions miss observations, like the cases in Africa and Asia. The publicly available high-quality data are very sparse and we will often encounter the extrapolation issue when predicting in these large ungauged regions. However, deep learning based methods with high flexibility and transferability provide new opportunities to tackle these issues by learning from data-rich continents and transferring the learned information. Our cross-continent PUR experiments are just designed for this purpose.

Line 236: Do you mean a median KGE of 0.78?

Yes, thanks for pointing this out. We are referring to the median KGE value of the 1675 subset catchments with long-term streamflow observations. We have modified this sentence as "*LSTM even reached a median KGE of 0.78*" to clarify it. {Lines 239-240}

Section 3.2: I expect dPL + evolved HBV with DP is always better than dPL + evolved HBV, because the former model is more flexible to capture the observation. If no better, it should

not be worse than the latter one. But Figure 5 shows, dPL + evolved HBV with DP is worse than dPL + evolved HBV in arid region. It will be helpful for the authors to clarify such Clarification.

The model with dynamic parameterization (dPL+evolved HBV with DP) is indeed more flexible than the other static parameterization model. That's why we always find the dynamic model has better fitting on the training data. However, Figure 5 is reporting the performance on the testing data that the models have never seen during the training. The dynamic model still performs better on the testing data in most cases, except for some specific cases like the temporal generalization in arid groups as the reviewer mentioned. These cases reflect another issue of overfitting. With more flexibility, the model also has a higher risk to be overfitted on the training dataset and reduce the generalization ability. This is why we only replace one or two parameters as dynamic ones to reduce this risk and also maintain physical clarity. In our previous paper we in fact warned against using too many dynamical parameters. In the CONUS dataset, the two-dynamic parameter model did not show more overfitting, presumably because the dataset was more homogeneous, while the behavior is somewhat different on this global dataset. This also shows that testing on a global dataset can give us different perspectives even on the similar issue. On the other hand, basins in the arid groups generally have lots of zero or very low flow rates, which makes these basins hard to simulate for all the models and very sensitive to the evaluation metrics. We use these statements in the main text to discuss these points:

*"The differentiable model with dynamic parameters performed better than the model with static parameters in all climate groups except the most challenging arid group. Dynamic parameterization with more structural flexibility generally provides stronger modeling ability, while also showing a higher risk of overfitting and degraded generalizability in basins which are very difficult to simulate."* {Lines 288-291}

Line 341: I don't think the median KGE = 0.58 is significantly better than median KGE = 0.52. They are pretty close performance to me. I suggest the authors to plot the cumulative density functions (CDFs) of the KGE for different model, and test if the CDFs are statistically different.

We performed the one-sided Wilcoxon signed-rank test and it shows that the KGE performance of the differentiable model is significantly better than the LSTM model (p-value less than 0.01). This can also be clearly seen from the box plot (Figure 6 in the main text) and the CDF plot below that the whole performance distribution of the differentiable model is leading that of the LSTM. We modified these statements to clarify this point, *"In the PUR scenario where European basins were held out for testing, differentiable models (median KGE=0.58) performed significantly better (p-value less than 0.01 using the one-sided Wilcoxon signed-rank test) than LSTM (median KGE=0.52)."* {Lines 348-350}

[Figure]

Line 350 – Line 351: Do you mean "cannot be obtained by straightforwardly training *physical* or *DL* models on data alone"? Figure 6 suggests the hybrid model is better than the traditional model calibration (Beck20). But it doesn't support that performance of hybrid model is statistically better than a purely data-driven model.

The differentiable modeling indeed achieves better extrapolation performance for the cross-continent PUR predictions, as shown in our response to the last comment. We modified this statement as below to avoid confusion.

"*With these results, we show that differentiable models have demonstrated a high simulation capability that cannot be obtained with traditional parameter regionalization approaches, and robust extrapolation capability in large data-sparse regions that is stronger than purely data-driven models like LSTM.*" {Lines 358-360}

Line 371: Why not using KGE and NSE for the ET evaluation to be consistent with discharge evaluation?

The remote sensing ET products usually have bias errors dependent on different algorithms. Correlation and RMSE are more commonly used metrics when evaluating ET variables (Velpuri et al., 2013; Holmes et al., 2018), which reflect the consistency of temporal dynamics and the average magnitude of differences between two datasets. We keep using correlation and RMSE as used in the previous studies so that we can provide context for our simulations at global scale.

Holmes, T. R., Hain, C. R., Crow, W. T., Anderson, M. C., & Kustas, W. P. (2018). Microwave implementation of two-source energy balance approach for estimating evapotranspiration. *Hydrology and earth system sciences*, *22*(2), 1351-1369.

Velpuri, N. M., Senay, G. B., Singh, R. K., Bohms, S., & Verdin, J. P. (2013). A comprehensive evaluation of two MODIS evapotranspiration products over the conterminous United States: Using point and gridded FLUXNET and water balance ET. *Remote Sensing of Environment*, *139*, 35-49.